# ANON: EXPLORING THE ADAPTIVITY OF OPTIMIZERS AND BEYOND

## ABSTRACT

Adaptive optimizers such as Adam have achieved great success in training large-scale models like large language models and diffusion models. However, they often generalize worse than non-adaptive methods, such as SGD on classical architectures like CNNs. We identify a key cause of this performance gap: *adaptivity* in pre-conditioners, which limits the optimizer's ability to adapt to diverse optimization landscapes. To address this, we propose **Anon** (**A**daptivity **N**on-restricted **O**ptimizer with **N**ovel convergence technique), a novel optimizer with **continuously tunable adaptivity** $\gamma \in \mathbb{R}$, allowing it to interpolate between SGD-like and Adam-like behaviors and even extrapolate beyond both. To ensure convergence across the entire adaptivity spectrum, we introduce *incremental delay update (IDU)*, a novel mechanism that is more flexible than AMSGrad's hard max-tracking strategy and enhances robustness to gradient noise. We theoretically establish convergence guarantees under both convex and non-convex settings. Empirically, Anon consistently outperforms state-of-the-art optimizers on representative image classification, diffusion, and language modeling tasks. These results demonstrate that adaptivity can serve as a valuable tunable design principle, and Anon provides the first unified and reliable framework capable of bridging the gap between classical and modern optimizers and surpassing their advantageous properties. Our code is available at https://anonymous.4open.science/r/Anon-6511/.

## 1 INTRODUCTION

Modern deep learning models rely heavily on optimization algorithms for effective training. Despite the wide success of adaptive optimizers such as Adam (Kingma & Ba, 2014) in large-scale models like diffusion networks (Nichol & Dhariwal, 2021; Rombach et al., 2022) and large language models (LLMs) (Brown et al., 2020; Touvron et al., 2023), they are often outperformed by non-adaptive methods such as SGD (Robbins & Monro, 1951) in classical architectures like CNNs (Wilson et al., 2017). These discrepancies raise a critical question: *Why do existing optimizers fail to generalize across diverse model families?*

We identify a key cause of this performance gap as ***adaptivity*** in pre-conditioners (i.e., the matrix that rescales the gradient before the step; SGD uses the identity, while Adam uses a data-dependent diagonal matrix). Whereas SGD applies fixed step sizes, adaptive optimizers such as Adam scale updates by gradient statistics, implicitly encoding an adaptivity level $A$ throughout training. This $A$, fixed without considering task-specific gradient distributions, can create a mismatch between the optimizer's adaptivity and the task's optimization landscape, potentially degrading generalization performance and rendering optimizers overly specialized. This motivates us to formalize and analyze adaptivity as a first-class property of optimizers.

To address this, we introduce a unified view of adaptivity, defined as the log-sensitivity of the pre-conditioner to global gradient scaling (§2.2). Existing optimizers correspond to fixed points on this adaptivity spectrum: SGD ($A = 0$), RMSProp (Graves, 2013) ($A \approx 1$), and Adam ($A \approx 1$). However, no method supports continuous control across $A \in \mathbb{R}$ with guaranteed stability.

We propose **Anon**, an **A**daptivity **N**on-restricted **O**ptimizer with **N**ovel convergence technique that enables *real-valued, tunable adaptivity* via a hyperparameter $\gamma \in \mathbb{R}$. Anon interpolates between SGD-like and Adam-like updates and even extrapolates beyond them. We note that such adaptivity comes with an important tradeoff: extreme adaptivity (e.g., $\gamma < 0$ or $\gamma > 1$) risks instability and

divergence. To tackle this tradeoff, we design a new convergence technique named **incremental delay update (IDU)**, which replaces hard max-tracking (e.g., in AMSGrad) with a soft, multi-scale accumulator that is provably stable.

**Our contributions are as follows**:

- We define a formal notion of adaptivity as a continuous control variable that unifies SGD, Adam, and beyond, offering a unifying lens to guide the design of future optimizers (§2.2).

- Through our analysis, we propose **Anon**, a novel universal optimizer which has tunable adaptivity. Anon's extensive range of adaptivity and adjustment endows the optimizer with the capability to surpass the performance ceiling inherent in previous optimizers. (§3.1).

- We propose a novel technique named *incremental delay update*, which eliminates the non-convergence risks in Anon arising from excessive range of adaptivity adjustment and anomalous negative adaptivity that may be set. We theoretically establish the convergence of Anon in both online convex and non-convex stochastic settings. In addition, we show that IDU can address convergence issues more effectively than AMSGrad's max-tracking approach. (§3.3).

- We conduct extensive experiments in image classification, language, and generative modeling, where Anon consistently outperforms strong baselines across tasks and architectures. (§4).

This work advocates for viewing adaptivity as a tunable principle and delivers the first provably stable, unified optimization framework that spans the full adaptivity spectrum.

## 2 PRELIMINARIES

### 2.1 REVIEW OF THE FRAME OF OPTIMIZERS

We focus on first-order optimizers, which are widely used to train deep learning models. To facilitate a unified understanding of their differences and commonalities, we introduce a generic framework, summarized in Algorithm 1. Here, $\mathcal{F}$ denotes the convex feasible set. $\boldsymbol{\theta} \in \mathcal{F}$ is the parameter to be optimal. Define $f(\boldsymbol{\theta})$ as a vector-valued function to minimize. $S_t$ is a diagonal matrix

---

**Algorithm 1:** Generic Optimizer Method Frame

**Input:** $\boldsymbol{\theta}, \eta, \{\phi_t, \psi_t\}_{t=1}^{\infty}$
**while** $\boldsymbol{\theta}_t$ not converged **do**
    $\boldsymbol{g}_t \leftarrow \nabla f_t(\boldsymbol{\theta}_t)$
    $\boldsymbol{m}_t \leftarrow (\phi_t(\boldsymbol{g}_{1:t,1}), ..., \phi_t(\boldsymbol{g}_{1:t,d}))^{\top}$
    $S_t \leftarrow \mathrm{diag}(\psi_t(\boldsymbol{g}_{1:t,1}), ..., \psi_t(\boldsymbol{g}_{1:t,d}))$
    $\boldsymbol{\theta}_t \leftarrow \Pi_{\mathcal{F},S_t}(\boldsymbol{\theta}_{t-1} - \eta(t)S_t^{-1}\boldsymbol{m}_t)$
**end while**

---

where $S_{t,i,i} := \psi_t(\boldsymbol{g}_{1:t,i})$. $\psi_t$ is the pre-conditioner function. $\prod_{\mathcal{F},S}(y) = \mathrm{argmin}_{x \in \mathcal{F}} \|S^{1/2}(x-y)\|$ denotes the projection of $y$ onto $\mathcal{F}$ under the scaling matrix $S$. The scheduler $\eta$ controls the learning rate at each step, which can be constant or scheduled via strategies such as cosine annealing (Loshchilov & Hutter, 2016). $\boldsymbol{g}_t$ is the gradient at step $t$. $\boldsymbol{m}_t$ is a vector where $\boldsymbol{m}_{t,i} := \phi_t(\boldsymbol{g}_{1:t,i})$. The momentum operator $\phi_t : \mathbb{R}^t \to \mathbb{R}$ is typically implemented as a moving average of past gradients. The two common variants are:

$$EMA(\boldsymbol{x}_{1:t}; \beta) = \frac{1-\beta}{(1-\beta^t)} \sum_{i=1}^{t} \beta^{t-i} x_i \, , \, M(\boldsymbol{x}_{1:t}; \beta) = \sum_{i=1}^{t} \beta^{t-i} x_i \, , \tag{1}$$

where *EMA* denotes the exponential moving average with bias correction. *M* refers to the classical momentum without normalization. Both operators serve to smooth the gradient history. **Since the smoothing behavior of $\phi$ is similar across optimizers, the key differentiator lies in the design of the pre-conditioner $\psi$.** Thus, we focus our subsequent analysis on the properties and effects of $\psi$.

While the momentum functions $\phi_t$ are largely similar across optimizers, the pre-conditioner functions $\psi_t : \mathbb{R}^t \to \mathbb{R}_+$ differ significantly and play a crucial role in shaping the optimizer's behavior. We summarize the designs of $\phi$ and $\psi$ for representative optimizers in Table 1.

As shown in Table 1, the momentum components $\phi$ exhibit similar behaviors across different optimizers. This observation highlights that the key distinction among optimizers arises from the design of $\psi$ rather than $\phi$. In fact, if we omit the bias correction factor $1/(1-\beta^t)$ in EMA, it effectively reduces to a classical momentum $M$ up to a constant scaling factor $1 - \beta$. Therefore, for the remainder of this paper, we primarily focus on analyzing the properties of the pre-conditioner $\psi$, assuming a shared momentum $\phi$ across optimizers unless otherwise noted.

Table 1: Summary of momentum functions and pre-conditioners for representative optimizers (Polyak, 1964; Luo et al., 2019; Zhuang et al., 2020). For full expressions of complex terms ($A_t^{\text{AMSGrad}}$, $A_t^{\text{AdaBound}}$, $A_t^{\text{AdaBelief}}$) $A_t^{\text{Anon}}$), please refer to Table 6 of Appendix B.1.

| Optimizer | $\phi_t(x)$ | $\psi_t(x)$ | $A_t(\psi, x)$ |
|---|---|---|---|
| SGD | $x_t$ | 1 | 0 |
| SGDM | $M(x; \beta)$ | 1 | 0 |
| RMSProp | $x_t$ | $\sqrt{EMA(x^2; \beta_2)} + \epsilon$ | $\frac{1}{1 + \epsilon/\sqrt{EMA(x^2; \beta_2)}}$ ($\approx 1$) |
| Adam | $EMA(x; \beta_1)$ | $\psi_t^{\text{RMSProp}}$ | $A_t^{\text{RMSProp}}$ |
| AMSGrad | $\phi_t^{\text{Adam}}$ | $\max_{i \in [t]}\{\psi_i^{\text{RMSProp}}\}$ | $[0, 1]$ ($\approx 1$) |
| AdaBound | $\phi_t^{\text{Adam}}$ | $\text{Clip}(\psi_t^{\text{RMSProp}}, f_l(t), f_u(t))$ | $[0, 1]$ ($1 \rightarrow 0$) |
| AdaBelief | $\phi_t^{\text{Adam}}$ | $\sqrt{EMA((x - \phi^{\text{Adam}})^2 + \epsilon/(1 - \beta_2); \beta_2)} + \epsilon$ | $[0, 1]$ ($\approx 1$) |
| Anon | $\phi_t^{\text{Adam}}$ | $\psi_t^{\text{Anon}}$ (equation 5) | $\approx \gamma$ |

Extensive empirical evidence has shown that SGD and SGDM often achieve better generalization than Adam in classical architectures such as ResNet (He et al., 2016), whereas Adam typically outperforms SGD in more complex architectures such as transformers. Understanding the fundamental causes behind this divergence remains an important question, with significant implications for the development of more effective optimizers. Several hypotheses have been proposed, including that Adam can escape saddle points more efficiently than SGD (Staib et al., 2019), and that SGD tends to find flatter minima whereas Adam is biased toward sharper minima, leading to superior generalization for SGD (Wilson et al., 2017). Regardless of the specific explanations, we hypothesize that the ultimate cause lies in how optimizers scale the loss landscape, a property we refer to as adaptivity. We will study how adaptivity affects optimization in § 3.2. Before that, we first give a formal definition of adaptivity.

## 2.2 THE ADAPTIVITY OF EXISTING OPTIMIZERS

We formalize the concept of adaptivity based on the framework described in Algorithm 1.

**Definition 1.** *Suppose the pre-conditioner $\psi_n$ is continuous. For any optimizer following Algorithm 1, we define the adaptivity $A$ of its pre-conditioner $\psi$ as*

$$A_n(\psi, x_{1:n}) = \nabla_k \ln \psi_n(k x_{1:n})\big|_{k=1}.$$

*Furthermore, we define two pre-conditioners $\psi$ and $\psi'$ are equivalent if and only if $A_n(\psi, x_{1:n}) = A_n(\psi', x_{1:n})$ for all $x_{1:n} \in \mathbb{R}^n$ and $n \in \mathbb{N}_+$.*

**Intuition Explanation** $A$ is a measure of the pre-conditioner's "response" to changes in the gradient's scale. $A = 0$ (like SGD) means the pre-conditioner is "non-reactive" and completely ignores the overall gradient scale. $A \approx 1$ (like Adam) means the pre-conditioner is "compensatory", adjusting itself with a strength of 1 to offset changes in gradient scale.

Notably, according to Definition 1, the adaptivity $A$ depends not only on the functional form of $\psi$, but also on the sequence of historical gradients $g_{1:t}$. This dependence reflects the fact that pre-conditioning is inherently dynamic: even for a fixed $\psi$, its adaptivity can vary during training as the distribution of gradients evolves. Separately, we introduce an important equivalence notion between pre-conditioners: even if two optimizers use different $\psi$ functions, they may be essentially equivalent from an adaptivity perspective.

**Theorem 1.** *If $\psi$ and $\psi'$ are from the same equivalence class, there is a function $f : \mathbb{N}_+ \to \mathbb{R}_+$ that makes $\psi_n(x_{1:n}) = \psi'_n(x_{1:n})f(n)$ for any $x_{1:n} \in \mathbb{R}^n$ and any $n \in \mathbb{N}_+$.*

**Decoupling from Scheduler** Theorem 1 shows that if two pre-conditioners yield the same adaptivity for any input, then they are equivalent. Specifically, if there exists a scheduler adjustment that can eliminate the difference between two pre-conditioners (e.g., $\psi' = k\psi$ corresponds to $\eta'(t) = k\eta(t)$), we regard them as equivalent strategies. The proof of Theorem 1 is deferred to Appendix E.

Based on these definitions, we can characterize the adaptivity of several widely used optimizers:

For SGD(M), the adaptivity is $A = 0$ in all dimensions, indicating no explicit scaling of the loss landscape. In contrast, for Adam and its variants (e.g., RMSProp, AdaBelief), the adaptivity is approximately $A = 1$, as the contribution of the small $\epsilon$ term is negligible compared to the accumulated gradient statistics most of the time. A more intricate case is AdaBound (Luo et al., 2019), whose adaptivity transitions dynamically from $A \approx 1$ toward $A \approx 0$ as training proceeds. Specifically, AdaBound clamps the pre-conditioner $\psi_t$ between shrinking bounds $\eta_l(t)$ and $\eta_u(t)$:

$$A_t(\psi^{\text{AdaBound}}, \boldsymbol{x}) = \begin{cases} A_t^{\text{RMSProp}}, & \text{if } \eta_l(t) < \psi_t^{\text{RMSProp}} < \eta_u(t), \\ 0, & \text{otherwise.} \end{cases} \quad (2)$$

As the bounds tighten over time, AdaBound behaves increasingly like SGD. This is supported by both evidence from Zhuang et al. (2020) and our experiments (Table 5), which indicates that AdaBound struggles in tasks such as GAN and diffusion model training, where high adaptivity is critical. These observations suggest the following: Optimizers with $A = 0$ (e.g., SGD) tend to generalize better on classical architectures such as CNNs, while those with $A = 1$ (e.g., Adam) perform better in complex modern architectures. However, whether $A = 0$, $A = 1$, or other values yield better performance remains an open question, which we explore in the next section.

## 2.3 THE OPTIMAL ADAPTIVITY FOR TASKS

We have observed that different tasks favor different levels of adaptivity $A$. This naturally raises a critical question: *Is $A = 0$ or $A = 1$ truly the optimal adaptivity for these tasks?*

As shown in Table 1, although mainstream adaptive optimizers typically have adaptivity close to 1, it is possible to adjust adaptivity by tuning hyperparameters such as $\epsilon$. For instance, by setting a large $\epsilon$ much greater than the accumulated moving average, the adaptivity of Adam and its variants can effectively approach 0. Indeed, prior works (Zaheer et al., 2018; Zhuang et al., 2020) have adopted this trick to align Adam's generalization performance more closely with SGD. Padam (Chen & Gu, 2018) offers another perspective by modifying the pre-conditioner as

$$\psi^{\text{Padam}} = (\psi^{\text{AMSGrad}})^{2p}, \quad A_t(\psi^{\text{Padam}}, \boldsymbol{x}) = \frac{2p}{1 + \epsilon / \max_{i \in [t]} \sqrt{EMA(\boldsymbol{x}_{1:i}^2; \beta_2)}}. \quad (3)$$

By adjusting $p \in [0, 0.5]$, Padam interpolates adaptivity between 0 and 1 while maintaining a small $\epsilon$. However, experiments from Chen & Gu (2018); Zhuang et al. (2020) show that Padam's performance typically lies between Adam and SGD, and only marginally surpasses them in limited scenarios. This observation raises a broader question: *Could adaptivity values beyond the $[0, 1]$ interval lead to even better performance?*

At first glance, one might attempt to extend adaptivity beyond $[0, 1]$ by simple functional modifications. However, expanding the adaptivity range is non-trivial. The convergence of most adaptive optimizers relies on the assumption:

$$\frac{\psi_t(g_{1:t+1,i})}{\eta(t+1)} \geq \frac{\psi_t(g_{1:t,i})}{\eta(t)}, \quad \forall i \in [d], \forall t \in \mathbb{N}_+, \quad (4)$$

which guarantees that the optimizer does not diverge even in the worst-case scenarios.

While in practice, the convergence condition is not strictly verified, optimizers like Adam typically exhibit stable behavior under standard training settings, suggesting that this assumption is likely satisfied. If we attempt to construct optimizers with negative adaptivity, new challenges arise. For example, setting $\psi = (\psi^{\text{Adam}})^\gamma$ with $\gamma < 0$ produces a negative adaptivity. However, setting the pre-conditioner to a negative power likely causes its value to decrease over time, thereby violating the critical convergence assumption. AMSGrad (Reddi et al., 2019) was introduced to address convergence issues inherent in Adam by enforcing a non-decreasing sequence in the denominator. Even with such safeguards, prior works (Chen & Gu, 2018; Chen et al., 2018) have shown that Padam, when extending adaptivity beyond $[0, 1]$, can still suffer from divergence in practice. Therefore, designing stable optimizers with tunable adaptivity beyond the classical range remains an open and challenging problem.

## 3 EXTEND TO ALL REAL NUMBERS

### 3.1 ADAPTIVITY TUNABLE OPTIMIZER AND BEYOND

In §2.2 and §2.3, we have shown that extending adaptivity beyond $[0, 1]$ could be beneficial. However, achieving tunable adaptivity across all real numbers while ensuring convergence remains challenging. We propose a new technique called *incremental delay update (IDU)*, which can ensure the convergence of an optimizer regardless of the value of its adaptivity. We will elaborate the technique in §3.3. Leveraging this technique, we design a novel optimizer *Anon* (**A**daptivity **N**on-restricted **O**ptimizer with **N**ovel convergence technique) with tunable adaptivity and extend the allowable range of adaptivity to all real numbers. The pseudocode of Anon is presented in Algorithm 2, and all the operations are element-wise. Here,

---

**Algorithm 2:** The Anon Optimizer

**Input:** $\eta$, $\beta_1$, $\beta_2$, $\epsilon$, $\gamma$
1 **Initialize** $\boldsymbol{\theta_0}$, $\boldsymbol{m_0} \leftarrow \boldsymbol{0}$ , $\boldsymbol{s_0} \leftarrow \boldsymbol{0}$, $t \leftarrow 0$, $k \leftarrow -1$
2 **while** $\boldsymbol{\theta_t}$ not converged **do**
3      $t \leftarrow t + 1$
4      $\boldsymbol{g_t} \leftarrow \nabla f_t(\boldsymbol{\theta_t})$
5      $\boldsymbol{m_t} \leftarrow \beta_1 \boldsymbol{m_{t-1}} + (1 - \beta_1)\boldsymbol{g_t}$
6      $\widehat{\boldsymbol{m}}_t \leftarrow \frac{\boldsymbol{m_t}}{1 - \beta_2^t}$
7      $\boldsymbol{s_t} \leftarrow \beta_2 \boldsymbol{s_{t-1}} + (1 - \beta_2)\boldsymbol{g_t^2}$
8      **if** $k + 1 = \log_2 t$ **do**
9          $k \leftarrow k + 1$
10          $\boldsymbol{\sigma_k} \leftarrow \boldsymbol{s_t}/(1 - \beta_2^{\max(t/2,1)}) + \epsilon$
11          $\boldsymbol{v_k} \leftarrow \sqrt{2/(\frac{1}{\boldsymbol{v_{k-1}^2}} + \boldsymbol{\sigma_k^\gamma})}$ if $k > 0$ else $\boldsymbol{\sigma_k^{-\gamma/2}}$
12          $\boldsymbol{s_t} \leftarrow \boldsymbol{0}$
13          $\boldsymbol{V_k} \leftarrow \text{diag}(v_{k,1}, ..., v_{k,d})$
14      **end if**
15      $\boldsymbol{\theta_t} \leftarrow \Pi_{\mathcal{F}, \boldsymbol{V_k^{-1}}}(\boldsymbol{\theta_{t-1}} - \eta(t)\boldsymbol{V_k}\widehat{\boldsymbol{m}}_t)$
16 **end while**

---

$\widehat{\boldsymbol{m}}_t$ corresponds to $\boldsymbol{m_t}$ in Algorithm 1. $\boldsymbol{V_k}$ corresponds to $\boldsymbol{S_t^{-1}}$ in Algorithm 1. $\boldsymbol{s_t}, \boldsymbol{\sigma_k}, \boldsymbol{v_k}$, and $k$ are intermediate variables. $\gamma$ is a hyperparameter to adjust adaptivity $A$. $\epsilon$ is a small hyperparameter to avoid division by 0. $\beta_1, \beta_2$ are hyperparameters for *EMA*, $0 \leq \beta_1, \beta_2 < 1$, typically set as 0.9 and 0.999. Let $\{a_n\}$ is a increasing sequence and $a_1 = 1$ (specially, let $a_0 = 0$). Let $\tilde{a}_n = \sum_{i>0} \mathbf{1}_{a_i \leq n}$, so $\tilde{a}_1 = 1$. The pre-conditioner of Anon can be written as equation 5 ($\beta_3 = 0.5, a_n = 2^{n-1}$):

$$\psi_t^{\text{Anon}}(\boldsymbol{x}) = \sqrt{\sum_{j=1}^{\tilde{a}_t} \beta_3^{\tilde{a}_t - j}(1 - \beta_3 \mathbf{1}_{j>1})EMA^\gamma(\boldsymbol{x}_{a_{j-1}+1:a_j}^2 + \epsilon; \beta_2)} \,. \tag{5}$$

**Theorem 2.** *For the optimizer Anon described in Algorithm 2, the adaptivity of Anon in $i$-th dimension is $\in [\gamma(1 - k), \gamma)$, where $k = \epsilon/\min_{j \in [\tilde{a}_t]} EMA(\boldsymbol{g}_{a_{j-1}+1:a_j, i}^2; \beta_2)$.*

According to Theorem 2, since we also set a small $\epsilon$ by default, we can adjust the adaptivity $A$ of Anon by adjusting the hyperparameter $\gamma$ ($A \approx \gamma$). The proof of Theorem 2 is shown in Appendix F.

### 3.2 How Adaptivity Influences Behaviors of Optimizers

**Empirical Validations** To show how adaptivity influences the behaviors of optimizers, we conduct a simple experiment in the loss function $f(x, y) = \ln(1 + \text{Beale}(x, y))/10$, where Beale (Beale, 1955) is a commonly used function to test optimizer performance. We apply appropriate learning rates for SGDM, Adam, AdaBelief, and Anon, and draw the optimization trajectories. We also show the loss landscapes in the view of Anon by scaling the loss landscape according to the pre-conditioner of Anon in epoch 100. The trajectories and loss landscapes after scaling are shown in Figure 1.

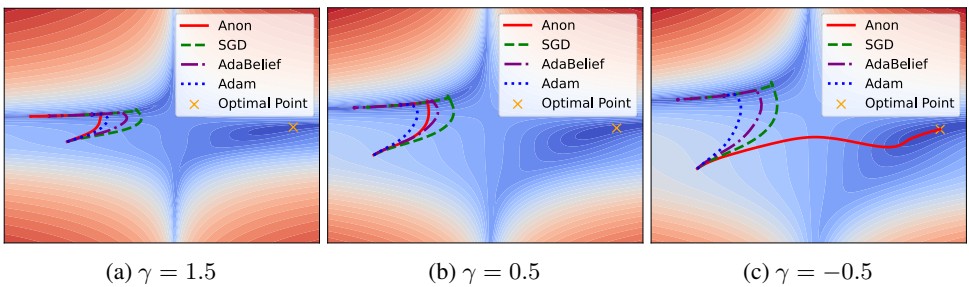

        (a) $\gamma = 1.5$                 (b) $\gamma = 0.5$                 (c) $\gamma = -0.5$

Figure 1: Trajectories of SGDM, Adam, AdaBelief, and Anon. The color change from deep red to deep blue represents the loss from high to low. And the loss landscape displayed is the result of scaling by Anon. More empirical experiments are shown in Appendix B.2 and D.

**Effect of Scaling** By changing $\gamma$ from 1.5 to $-0.5$, the adaptivity also changes from 1.5 to $-0.5$ referring to Theorem 2. We can find that when $\gamma = 1.5$, Anon takes a shorter path to descend along the y-axis. When $\gamma = 0.5$, the path is between Adam and SGDM. And when $\gamma = -0.5$, the Anon descends along the x-axis and arrives at the optimal point. We can find that in the progress of $\gamma$'s decreasing, the scale of the x-axis is smaller and smaller than that of the y-axis, so that Anon can choose the right path to reach the optimal point. This example implies that the optimization path of Anon in deep learning training may be greatly different from other optimizers, helping reach a new parameter region that makes the model achieve better performance.

**The Meaning of Negative Adaptivity** Positive adaptivity typically reduces step sizes for large gradients to help escape saddle points. In contrast, negative adaptivity adopts the opposite strategy by increasing step sizes when gradients are large, which enables the optimizer to escape from sharp minima. Intuitively, higher adaptivity drives the optimization toward steeper minima, whereas lower adaptivity favors flatter regions. **Thus, adaptivity influences the optimizer not only through its path but also by altering its preference for specific minima geometries.** This perspective implies that restricting adaptivity to fixed points like A=0 or A=1 is insufficient. Empirically, we find that negative adaptivity is more effective for classical models, while positive adaptivity remains suitable for complex architectures.

### 3.3 INCREMENTAL DELAY UPDATE

As we state in § 2.3, it is challenging to guarantee the convergence when adaptivity is allowed to take any value. So we propose a new technique *incremental delay update* (IDU), which can be seen as using a new function $U(\boldsymbol{x}; \psi^{\text{old}})$ to replace the old pre-conditioner function $\psi$:

$$U_t(\boldsymbol{x}; \psi_t^{\text{old}}, \{a_n\}, \beta_3) = \sqrt{\sum_{j=1}^{\tilde{a}_t} \beta_3^{\tilde{a}_t - j} \left(1 - \beta_3 \mathbf{1}_{j>1}\right) \left(\psi_{a_j - a_{j-1}}^{\text{old}}(\boldsymbol{x}_{a_{j-1}+1:a_j})\right)^2}. \quad (6)$$

Line 9~15 of Algorithm 2 are the recursive formulas for IDU used in Anon where $\beta_3 = 0.5$, $a_n = 2^{n-1}$ and $\psi^{\text{old}} = EMA^\gamma(\boldsymbol{x}^2 + \epsilon; \beta_2)$. IDU updates the pre-conditioner using accumulated gradient information **only at specific, delayed steps**. This strategy confines unpredictable oscillations within a manageable range, thereby ensuring theoretical convergence while still permitting the pre-conditioner to change non-monotonically. We show the convergence of Anon in Theorem 3 (convex cases) and Theorem 4 (non-convex cases). And the proofs are provided in Appendix G and H.

**Theorem 3.** *(Convergence analysis for online convex optimization) Let $\{\theta_t\}$ and $\{v_k\}$ be the sequence obtained by Algorithm 2, $\gamma \in \mathbb{R}$, $\beta_1 \in [0,1)$, $\beta_2 \in [0,1)$, $\beta_{1,t+1} \in [0,\beta_1]$, $\beta_{1,1} = \beta_1$, $\eta(t) = \frac{\eta_0}{\sqrt{t}}$, for $\forall t \in [T]$. Assume that $\|x - y\|_\infty \leq D_\infty$ for $\forall x, y \in \mathcal{F}$. Suppose $f(\theta)$ is a convex function, $\|g_t\|_\infty \leq G_\infty$, for $\forall t \in [T]$, $\theta \in \mathcal{F}$. Let $C_l = \min(G_\infty^{-\gamma}, \epsilon^{-\gamma})$, $C_u = \max(G_\infty^{-\gamma}, \epsilon^{-\gamma})$, where $\epsilon \in \mathbb{R}_+$ is a very number set in Algorithm 2. The optimal point of $f$ is denoted as $\theta^*$. For $\{\theta_t\}$ generated by Anon, there is a bound on the regret:*

$$\sum_{t=1}^T [f_t(\theta_t) - f_t(\theta^*)] \leq \frac{dD_\infty^2 c_l^{-1}}{(1-\beta_1)\eta_0} \left(\sqrt{T} + \sum_{k=1}^{\tilde{a}_T - 1} \sqrt{a_{k+1}}\right) + \sum_{t=1}^{T-1} \left[\frac{\beta_{1,t+1} \mathbf{1}_{\beta_{1,t+1} > \beta_{1,t}} D_\infty^2}{2C_l \eta_{t+1}(1-\beta_1)^2}\right]$$

$$+ \frac{D_\infty^2}{2C_l \eta_1(1-\beta_1)} + \frac{dD_\infty G_\infty}{1-\beta_1} \sum_{t=1}^T \beta_{1,t} + \frac{dG_\infty^2 C_u \eta_0}{1-\beta_1}\sqrt{T} \quad (7)$$

**Corollary 3.1.** *Suppose $\beta_{1,t} = \beta_1 \lambda^t$, $0 < \lambda < 1$ in Theorem 3, then we have:*

$$\sum_{t=1}^T [f_t(\theta_t) - f_t(\theta^*)] \leq \frac{dD_\infty^2 c_l^{-1}}{(1-\beta_1)\eta_0} \left(\sqrt{T} + \sum_{k=1}^{\tilde{a}_T - 1} \sqrt{a_{k+1}}\right) + \frac{D_\infty^2}{2C_l \eta_1(1-\beta_1)}$$

$$+ \frac{dD_\infty G_\infty \beta_1}{(1-\beta_1)(1-\lambda)} + \frac{dG_\infty^2 C_u \eta_0}{1-\beta_1}\sqrt{T} \quad (8)$$

It implies the regret of Anon is upper-bounded by $O(\sqrt{T})$ for convex case when $a_n = 2^{n-1}$.

**Theorem 4.** *(Convergence analysis for non-convex stochastic optimization) The update of $\theta_t$ can be described as $\theta_{t+1} = \theta_t - \eta_t V_{\lfloor \log_2 t \rfloor} m_t$, and $m_t = \beta_1 m_{t-1} + (1 - \beta_1) g_t$.*

*Under the assumptions:*

- *$f$ is differentiable and $f^* \leq f \leq F$. $\nabla f(x)$ is L-Lipschitz continuous, i.e. $\|\nabla f(x) - \nabla f(y)\| \leq L\|x - y\|$, $\forall x, y$.*
- *The noisy gradient is unbias and its infinity norm is bounded by N, i.e. $\mathbb{E}g_t = \nabla f(x)$, $\|g_t\|_\infty \leq N$.*

*The hyperparameters are set as: $\eta_t = \eta_0 t^{-p}$, $\eta_0 > 0$, $p \in (0, 1)$ where the bounds are $C_l I \preceq V_{\lfloor \log_2 t \rfloor} \preceq C_u I$, and $0 < C_l < C_u$ ($A \preceq B$ means $B - A$ is a positive semi-definite matrix). And $\epsilon$ and $N$ ensure $C_l$ and $C_u$ exist. For sequence $\{\theta_t\}$ generated by Anon, we have:*

$$\frac{1}{T} \sum_{t=1}^{T} \left\| \nabla f(x_t) \right\|^2 \leq \frac{1}{\eta_0 C_l} T^{p-1} \left( F - f^* + K \int_1^T t^{-2p} \, \mathrm{d}t + J + K \right), \tag{9}$$

*where*

$$J = \frac{\beta_1^2 d}{4L(1-\beta_1)^2} N^2 + \frac{3dN^2}{1-\beta_1} \eta_0 C_u \sum_{k=1}^{\tilde{a}_t} (a_k - \mathbf{1}_{k \neq 1})^{-p}, \quad K = \left( \frac{1}{1-\beta_1} + \frac{1}{2} \right) L \eta_0^2 N^2 C_u^2 d$$

Theorem 4 shows when $p = 0.5$ and $a_n = 2^{n-1}$, Anon has a convergence rate of $O(\ln T / \sqrt{T})$ for non-convex cases. Note that the convergence rates shown in Theorem 3 and Theorem 4 are the same as mainstream adaptive optimizers under the strong assumption equation 4 or using the technique of AMSGrad. And the assumptions and boundedness conditions are standard in the literature and consistent with those adopted in previous works like Luo et al. (2019) and Zhuang et al. (2020).

**Better Noise Robustness** Other convergence guarantee techniques typically employ alternative methods to ensure equation 4 holds, thereby guaranteeing optimizer convergence. Noise in the early training stage can greatly influence their performance, making it difficult for these methods to use the information of the latest gradients. As we know, IDU is the first technique that makes optimizers converge and allows equation 4 to not hold, which will offer Anon (IDU) better noise robustness and flexibility. To evaluate the robustness of IDU against noise, we do further experiments where we compare Anon (IDU) and AMSGrad. Slightly different from the Table 1, AMSGrad is usually implemented in practice in the form: $\max_{i \in [t]} \{\psi_i^{\text{RMSProp}} \sqrt{1 - \beta_2^i}\} / \sqrt{1 - \beta_2^t}$ (we apply in experiments). But regardless of the first form or the second form, we can extrapolate that AMSGrad's strategy of persistently applying the max operation is highly susceptible to noise interference. We conduct empirical experiments to prove it, and the relevant function settings include:

$$f_t(x) = \begin{cases} 1010x, & \text{if } t \bmod 101 = 1 \\ -10x, & \text{otherwise} \end{cases}, N_t = \begin{cases} 500/\mathrm{e}^{t-1}, & \text{if } t \bmod 2 = 1 \\ -500/\mathrm{e}^{t-1}, & \text{otherwise} \end{cases} \tag{10}$$

with the constraint set $\mathcal{F} = [-1, 1]$. The $f_t(x)$ is the example provided in Reddi et al. (2019), which can make Adam diverge. And $N_t$ is the noise added to the gradients $g_t$. We can observe that the noisy gradient is unbiased and its influence on gradients approaches $0$ with the increase of $t$. The results of experiments are shown in Figure 2. Note that we set $\gamma = 1$ to make the adaptivity of Anon equivalent to AMSGrad and Adam, and their other hyperparameters are the same. Therefore, we can compare the performances of the two convergence guarantee techniques fairly.

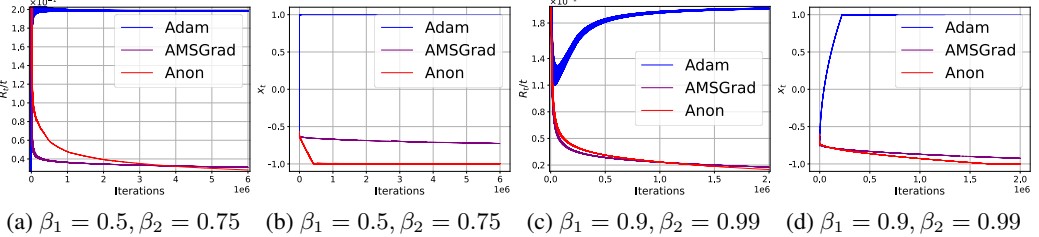

(a) $\beta_1 = 0.5, \beta_2 = 0.75$    (b) $\beta_1 = 0.5, \beta_2 = 0.75$    (c) $\beta_1 = 0.9, \beta_2 = 0.99$    (d) $\beta_1 = 0.9, \beta_2 = 0.99$

Figure 2: Comparison of Adam, AMSGrad, and Anon on a simple convex problem with noise. The setting of hyperparameters follows $\beta_1 < \sqrt{\beta_2}$ and $\eta(t) = 0.1/\sqrt{t}$ (Reddi et al., 2019).

From Figure 2(a)(c), we can see that the regrets divided by $t$ of Anon and AMSGrad approach $0$ gradually, meaning they converge. And those of Adam approach a constant, meaning it diverges.

Although both Anon and AMSGrad can converge, Figure 2(b)(d) shows that Anon can reach the optimal point $x = -1$ fast, but AMSGrad converges to the optimal point much slower due to the noise, especially when $\beta_2$ is small. The result proves that Anon (IDU) has better noise robustness than AMSGrad, as we have inferred. It forms the theoretical backbone of Anon and opens new avenues for designing flexible optimizers.

# 4 EXPERIMENTS

In this section, we compare Anon with 13 baseline optimizers, including SGD(M), Adam, AdamW (Loshchilov & Hutter, 2017), Yogi (Zaheer et al., 2018), AdaBound, RAdam (Liu et al., 2019), SWA (Izmailov et al., 2018), Lookahead (Zhang et al., 2019), AdaBelief, Adai (Xie et al., 2022) Lookaround (Zhang et al., 2023), Sophia (Liu et al., 2023), AGD (Yue et al., 2023) and HVAdam (Zhang et al., 2025) by validating Anon in various tasks including image classification tasks on ResNet, image generation on diffusion model and natural language processing tasks on LLMs. Except for experiments on the diffusion model, all the benchmarks are from the data presented in the paper. Therefore, the hyperparameters of other optimizers have been extensively searched.

**Image Classification with CNN** We conduct experiments on ImageNet (Russakovsky et al., 2015) with ResNet18 and ResNet50. We use the official implementation of AdaBound, AdaBelief and Lookaound, so the replication is exact. For ResNet50, the top-1 accuracy is reported in Table 3. And for ResNet18, the top-1 accuracy is shown in Table 2. We set 1 learning rate for Anon, which corresponds to 0.1 learning rate and 0.9 momentum setting of SGDM, because $EMA(\boldsymbol{x}; 0.9) \approx M(\boldsymbol{x}; 0.9)/10$ according to equation 1. We set $\gamma = -0.1$ for Anon ($A = -0.1$), and it surpasses the performance of SGDM ($A = 0$). These results prove our guess that the negative adaptivity is more suitable for classical models like CNNs.

Table 2: Top-1 accuracy (%) of ResNet18 on ImageNet. † from Chen & Gu (2018), ‡ from Liu et al. (2019), ∗ from Zhuang et al. (2020).

| **Anon** | SGDM | AMSGradW | AdaBelief | AdaBound[†] | Yogi[†] | Adam[‡] | MSVAG[∗] | RAdam[‡] |
|---|---|---|---|---|---|---|---|---|
| **70.06** | 69.94 | 68.78 | 69.42 | 68.13 | 68.23 | 66.54 | 65.99 | 67.62 |

Table 3: Top-1 accuracy (%) of ResNet50 on ImageNet. † from Xie et al. (2022), ‡ from Zhang et al. (2023), ∗ from Zhang et al. (2025).

| **Anon** | SGDM | Lookaround | Adam[†] | Adai[†] | SWA[‡] | Lookahead[‡] | HVAdam[∗] |
|---|---|---|---|---|---|---|---|
| **77.25** | 76.23 | 76.77 | 72.87 | 76.80 | 76.78 | 76.52 | 77.22 |

**Language Modeling** We train autoregressive models on OpenWebText (Gokaslan & Cohen, 2019) using the official implementation of Sophia (Liu et al., 2023). Our experiments follow the exact experimental setup and hyperparameter configurations of Liu et al. (2023). We set $\gamma \geq 1$ and use other optimizers' learning rate setting for Anon. The results of experiments

Table 4: Validation loss and training time on OpenWebText.

| Model | Optimizer | Validation Loss | Time (h) |
|---|---|---|---|
| GPT2-small | Anon$_{\gamma=1.1}$ | **2.93283** | **26.17554** |
| | AdamW | 2.95614 | 26.88118 |
| | Sophia-G | 2.95143 | 28.98702 |
| GPT2-medium | Anon$_{\gamma=1}$ | **2.69017** | 36.91487 |
| | AdamW | 2.70994 | **36.83633** |
| | Sophia-G | 2.70653 | 41.02486 |

are presented in Table 4, and Anon obtains the lowest validation losses in GPT2-small and GPT2-medium, demonstrating strong performance on LLM training. Note that through our experiments, we find that many variants of Adam are slower than Adam because they introduce extra calculations. But from Table 4 we can see that Anon obtains the compared and even faster speed than Adam. This is because when iterations approach infinity, for the average time cost per iteration, we have

$$E(t^{\text{Adam}} - t^{\text{Anon}}) \approx t^{vector\text{-}Div} + t^{vector\text{-}Sqrt} - t^{vector\text{-}Mul} - C\frac{\log_2 Iters}{Iters} > 0 \ (Iters \rightarrow \infty), \quad (11)$$

and $C$ is the time cost of the operations in line 9~15 of Algorithm 2 per iteration. From equation 11 we can find that the Adam's time cost of per iteration is more than Anon's, since the vector division is slower than vector multiplication. Furthermore, IDU makes the big time cost of vector power operation related to $\gamma \in \mathbb{R}$ used in Anon (covered in $C$) approach 0, which greatly improved the practical value of Anon.

**Image Generation with Diffusion Model**   We conduct image generation experiments on CIFAR-10 (Krizhevsky et al., 2009) with diffusion model.   We search the learning rate in $\{0.1, 0.01, 0.001, 0.0001, 0.00001\}$ for AdamW, AMSGrad, Anon, SGDM, and AdaBound. The code and the settings of other hyperparameters are consistent with the official implementation of Nichol & Dhariwal (2021). The results are reported in Table 5. When set learning rate 0.0001 (also the most suitable value for Adam) and $\gamma = 1.01$, Anon achieves SOTA and proves that the adaptivity higher than 1 is a better choice for complex models.

Table 5: FID scores of diffusion models on CIFAR-10 (lower is better).

| Adam | AMSGrad | SGDM | AdaBound | Anon$_{\gamma=1}$ | Anon$_{\gamma=1.01}$ |
|---|---|---|---|---|---|
| 9.11 | 8.12 | 12.84 | 12.13 | 8.03 | **7.75** |

**Comprehensive Analysis and Robustness**   From the results on CNNs, we observe that setting the learning rate corresponding to SGDM and applying a negative adaptivity leads to better generalization and higher accuracy. In contrast, setting the learning rate equivalent to Adam and using a positive adaptivity ($\gamma \geq 1$) achieves SOTA results in diffusion models and LLMs. This observation aligns well with our analysis in Section 2.3, highlighting that adaptivity is a key factor in model-specific optimizer behavior. Additionally, our results demonstrate the practical benefits of the proposed IDU mechanism in improving training efficiency: it accelerates computation by transforming expensive operations into negligible cost as shown in equation 11, and this benefit can extend to other optimizers as well. We also show the FID of setting of $\gamma = 1$ (the same as Adam) in

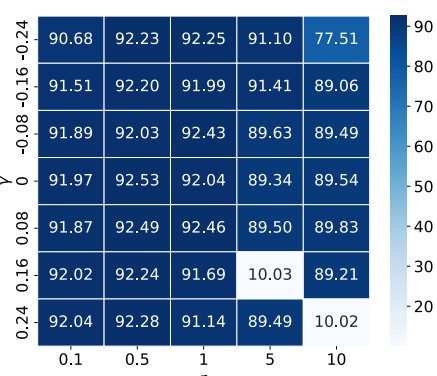

Figure 3: Hyperparameter sensitivity analysis of ResNet20 on CIFAR-10

Table 5 and Table 4 which means the only difference is the inclusion of IDU in Anon, and it also outperforms other optimizers, presenting the **improvement brought by IDU**. Furthermore, we assess the robustness of Anon to hyperparameter choices. As illustrated in Figure 3, Anon maintains high performance across a broad range of learning rates and $\gamma$ values. Notably, unlike many adaptive optimizers that require tuning of $\beta_1$, $\beta_2$, and $\epsilon$ per task, we use fixed settings ($\beta_1 = 0.9$, $\beta_2 = 0.999$, $\epsilon = 10^{-16}$) throughout all experiments. Despite this, Anon consistently achieves SOTA, validating its robustness and the practical applicability of our proposed design.

## 5   CONCLUSION

We propose Anon, a novel optimizer that obtains tunable non-restricted adaptivity and IDU convergence guarantee technique. The results of deep learning experiments show that Anon outperforms almost all other optimizers, which demonstrates the superiority of Anon and verifies the correctness of our idea about adaptivity. And we prove that Anon's convergence rate in both convex and nonconvex cases can achieve the convergence rate of mainstream optimizers under the strong assumption or with AMSGrad's technique. And the experimental results and theoretical analysis show IDU matches AMSGrad's convergence rate and memory cost. In addition, IDU offers better noise robustness, more flexibility, and even accelerates certain operations in practice. Therefore, we believe that IDU is overall superior to the convergence technique of AMSGrad. And follow the settings of those original papers, the experiments use many techniques like cosine annealing, decoupled weight decay regularization, and gradient clipping by default, so it means Anon is perfectly compatible with these widely used techniques. Thus, we expect Anon can become the preferred optimizer in extensive fields of deep learning due to its great performance.

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

# APPENDIX

## A    LIMITATION AND FUTRUE WORK

Although we prove that the adaptivity is an important attribute for first-order optimizers, there are a small number of first-order optimizers not covered by our Adaptivity Definition 1 such as HVAdam which does not conform to the frame outlined in Algorithm 1. For this situation, we will try to give a more general adaptivity definition in the future. And limited by computational resources, our hyperparameter search for Anon was incomplete. For example, in diffusion model trials, a learning rate of $0.0001$ with adaptivity $1.02$ caused the early training loss hard to decrease, whereas a learning rate of $10^{-5}$ allowed higher adaptivity such as $1.15$. Regrettably, time constraints prevented further exploration of these observations so further investigation is needed to fully explore Anon's potential. We also hope this work can contribute to exploring the design of deep learning models, as our experiments reveal distinct adaptivity preferences across different model architectures. This observation suggests that certain "ineffective" modifications proposed for neural networks might simply stem from usual optimal adaptivity (i.e., values deviating from the conventional $[0, 1]$ range), rather than inherent flaws in the design concept. In such scenarios, Anon's extensive adaptivity tuning capacity could potentially unlock the latent capabilities of these architectures.

## B    ADAPTIVITY OF OPTIMIZERS

### B.1    THE ADAPTIVITY OF OPTIMIZERS

We present the full adaptivity table of some optimizers mentioned in the main paper in Table 6.

Table 6: Summary of adaptivity for representative optimizers.

| Optimizer | $A_t(\psi, x)$ |
|:---:|:---:|
| SGD | $0$ |
| SGDM | $0$ |
| RMSProp | $\dfrac{1}{1+\epsilon/\sqrt{EMA(x^2;\beta_2)}}$ |
| Adam | $A_t^{\text{RMSProp}}$ |
| AMSGrad | $\dfrac{1}{1+\epsilon\big/\max_{i\in[t]}\sqrt{EMA(x_{1:i}^2;\beta_2)}}$ |
| Padam | $\dfrac{2p}{1+\epsilon\big/\max_{i\in[t]}\sqrt{EMA(x_{1:i}^2;\beta_2)}}$ |
| AdaBound | $\begin{cases} A_t^{\text{RMSProp}}, & \text{if } \eta_l(t) < \psi_t^{\text{RMSProp}} < \eta_u(t), \\ 0, & \text{otherwise.} \end{cases}$ |
| AdaBelief | $\dfrac{1}{1+\epsilon\cdot\left[\frac{1}{1-\beta_2}+\sqrt{EMA((x-\phi^{\text{Adam}})^2+\epsilon/(1-\beta_2);\beta_2)}\right]/EMA((x-\phi^{\text{Adam}})^2;\beta_2)}$ |
| Anon | equation 13 ($\approx \gamma$) |

For Algorithm 2, we provide an equivalent formulation that, while yielding no speedup, offers a clearer representation of its underlying mechanism in Algorithm 3.

---

**Algorithm 3:** The Anon Optimizer

---

1 **Input:** $\eta, \beta_1, \beta_2, \epsilon, \gamma$
2 **Initialize $\theta_0$, $m_0 \leftarrow \mathbf{0}$, $s_0 \leftarrow \mathbf{0}$, $t \leftarrow 0$, $k \leftarrow 0$, $a \leftarrow 0$**
3 **while $\theta_t$ not converged do**
4     $t \leftarrow t + 1$
5     $g_t \leftarrow \nabla f_t(\theta_t)$
6     $m_t \leftarrow \beta_1 m_{t-1} + (1 - \beta_1) g_t$
7     $\widehat{m}_t \leftarrow \frac{m_t}{1 - \beta_2^t}$
8     $s_t \leftarrow \beta_2 s_{t-1} + (1 - \beta_2) g_t^2$
9     **if $t = 2^k$ do**
10         $\sigma_k \leftarrow s_t / (1 - \beta_2^{2^k - a}) + \epsilon$
11         $a \leftarrow 2^k$
12         $v_k \leftarrow \frac{v_{k-1}^2 + \sigma_k^\gamma}{2}$ if $k > 1$ else $\sigma_k^\gamma$
13         $s_t \leftarrow \mathbf{0}$
14         $V_k \leftarrow \text{diag}(\sqrt{v_{k,1}}, ..., \sqrt{v_{k,d}})$
15         $k \leftarrow k + 1$
16     **end if**
17     $\theta_t \leftarrow \Pi_{\mathcal{F}, V_{k-1}}(\theta_{t-1} - \eta(t) V_{k-1}^{-1} \widehat{m}_t)$
18 **end while**

---

## B.2   The Effect of Adaptivity

To intuitively illustrate the impact of adaptivity, we present a visualization in Figure 4.

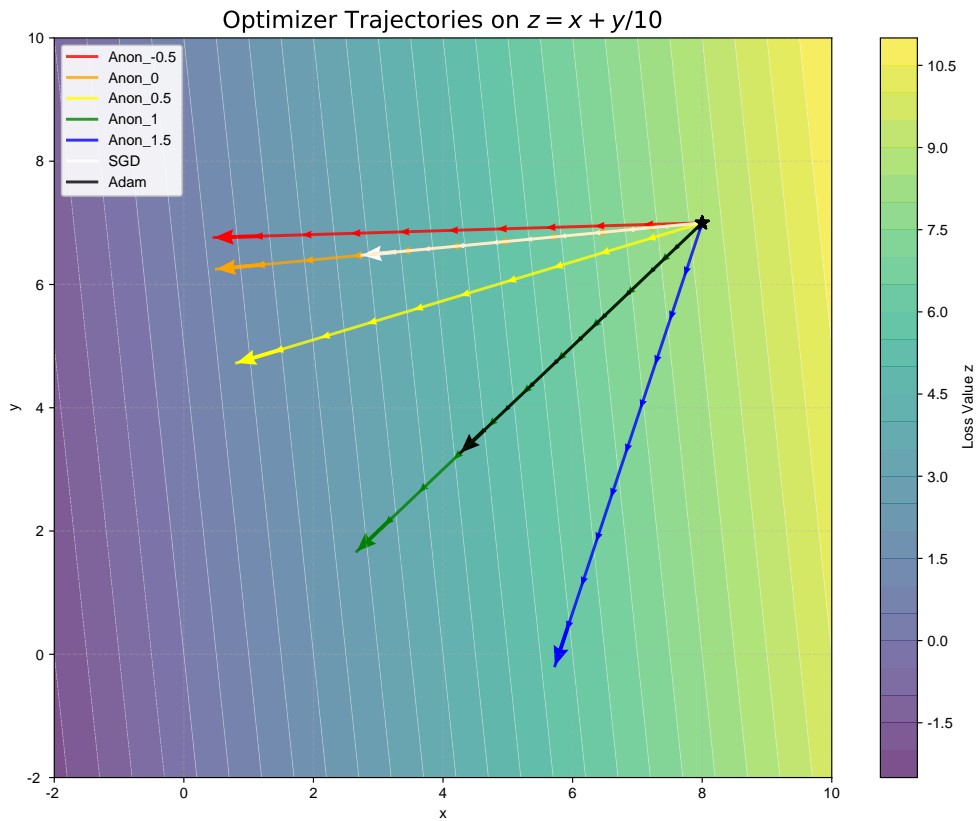

Figure 4: Optimization trajectories of SGDM, Adam, and Anon with varying $\gamma$. The gradient from yellow to purple indicates decreasing loss values. Different learning rates are applied to clearly visualize the distinct update directions.

We further demonstrate that varying adaptivity drives optimizers toward distinct regions of the parameter space by training GPT2-small on OpenWebText. The validation loss results are presented in Table 7. Additionally, we analyze the cosine similarity of the trained model parameters in Table 8. Notably, Anon with $\gamma = 1.15$ exhibits the lowest similarity when compared to Adam, Lion (Chen et al., 2023), and Muon (Jordan et al., 2024), suggesting it discovers a unique solution.

Table 7: Validation loss on OpenWebText.

|      | $\text{Anon}_{\gamma=1}$ | $\text{Anon}_{\gamma=1.1}$ | $\text{Anon}_{\gamma=1.15}$ | Adam | Lion | Muon |
|------|------|------|------|------|------|------|
| Loss | 2.937 | **2.927** | 2.932 | 2.934 | 2.992 | 3.092 |

Table 8: Cosine similarity on OpenWebText. The upper number in each cell represents the cosine similarity of **all parameters**, while the lower number represents the cosine similarity of the **weights only**.

|  | $\text{Anon}_{\gamma=1}$ | $\text{Anon}_{\gamma=1.1}$ | $\text{Anon}_{\gamma=1.15}$ | Adam | Lion | Muon |
|---|---|---|---|---|---|---|
| $\text{Anon}_{\gamma=1}$ | 1
1 | 0.597
0.317 | 0.355
0.201 | 0.914
0.511 | 0.857
0.161 | 0.901
0.194 |
| $\text{Anon}_{\gamma=1.1}$ | 0.597
0.317 | 1
1 | 0.425
0.328 | 0.573
0.248 | 0.522
0.088 | 0.539
0.098 |
| $\text{Anon}_{\gamma=1.15}$ | 0.335
0.201 | 0.425
0.328 | 1
1 | 0.335
0.156 | 0.299
0.056 | 0.306
0.063 |
| Adam | 0.914
0.511 | 0.573
0.248 | 0.355
0.156 | 1
1 | 0.880
0.191 | 0.923
0.222 |
| Lion | 0.857
0.161 | 0.522
0.088 | 0.299
0.056 | 0.880
0.191 | 1
1 | 0.925
0.132 |
| Muon | 0.901
0.194 | 0.539
0.098 | 0.306
0.063 | 0.923
0.222 | 0.925
0.132 | 1
1 |

## C  DETAILS OF EXPERIMENTS AND MORE EXPERIMENTS

### C.1  IMAGE CLASSIFICATION

**ResNet20 and ResNet32**  We also do experiments on CIFAR-10 (Russakovsky et al., 2015) with ResNet20 and ResNet32 and achieve the SOTA. The results are presented in Table 9 (all other optimizers' data is from Yue et al. (2023)), and the detailed setting is shown in Appendix C. We report the results of all other optimizers from AGD (Yue et al., 2023) and adopt the same experimental setup as in the official implementation[1]. And do hyperparameters searching for Anon as Figure 3 in the main paper ( $\eta \in [0.1, 10]$, $\gamma \in [-0.24, 0.24]$ ) and finally select $\eta = 1$, $\gamma = -0.08$ for ResNet20 and $\eta = 0.5$, $\gamma = -0.17$ for ResNet32. Like the default setting for AdamW, AGD and AdaHessian (Yao et al., 2021) in the two experiments, we use the decoupled weight decay for Anon.

Table 9: Top-1 accuracy(%) comparison on CIFAR-10 (ResNet models)

| Model | Optimizers | | | | | | |
|---|---|---|---|---|---|---|---|
|  | SGD | Adam | AdamW | AdaBelief | AdaHessian | AGD | **Anon** |
| ResNet20 | 92.14±.14 | 90.46±.20 | 92.12±.14 | 92.19±.15 | 92.27±.27 | 92.35±.24 | **92.47±.05** |
| ResNet32 | 93.10±.07 | 91.54±.12 | 92.72±.01 | 92.90±.13 | 92.91±.14 | 93.12±.18 | **93.20±.08** |

**ResNet18**  We report the results from the sources stated in the main paper. We adopt the same experimental setup as in the official implementation[2], and reproduce the results of SGDM, AdaBelief under the official recommended hyperparameter setting. We search learning rate in {0.1, 0.01, 0.001 } for AMSGrad with decoupled weight decay, and the best value is 0.01. We set learning rate as 1 and search $\gamma$ in {-0.1, -0.05, 0, 0.05} for Anon and the best value is -0.1.

**ResNet50**  We report the results from the sources stated in the main paper. We adopt the same experimental setup as in the official implementation[3], and reproduce the results of SGDM, LookAround under the official recommended hyperparameter setting. Due to the heavy calculation burden, we do not do much searching and simply set $\eta = 1$ and $\gamma = -0.1$ for Anon.

---

[1]https://github.com/amirgholami/adahessian

[2]https://github.com/juntang-zhuang/Adabelief-Optimizer

[3]https://github.com/Ardcy/Lookaround

## C.2 IMAGE GENERATION

**Diffusion Model** We adopt the same experimental setup as in the official implementation[4] (Unconditional CIFAR-10 with L_hybrid objective and cosine noise schedule). And search learning rate in {0.1, 0.01, ... , 0.00001} for all optimizers and search $\gamma$ in {1, 1.1, 1.01} for Anon. The optimal choice is $\eta = 0.0001$ and $\gamma = 1.01$.

## C.3 LANGUAGE MODELING

**GPT2** We refer to the experimental setup in the official implementation[56] and set nproc_per_node=4 due to limited computational resources. Under this setting, we find that when apply the same learning rate scheduler as Sophia in GPT2-medium, AdamW can get lower loss, so we apply this new setting for AdamW and Anon. We set $\gamma = 1$ for Anon. And all the optimizers use decoupled weight decay.

## C.4 ABLATION STUDY ON IDU HYPERPARAMETERS

We conduct an ablation study to investigate the impact of the hyperparameters $\{a_n\}$ and $\beta_3$ in IDU. The experiments are performed using ResNet20 on the CIFAR-10 dataset, with results summarized in Table 10. **These results demonstrate that IDU is robust to hyperparameter variations; indeed, certain configurations (e.g., $\beta_3 = 0.3, a_n = 4^{n-1}$) even outperform our default setting ($\beta_3 = 0.5, a_n = 2^{n-1}$).**

Table 10: Ablation study on the hyperparameters $\{a_n\}$ and $\beta_3$ of IDU.

|  | $\beta_3 = 0.1$ | $\beta_3 = 0.3$ | $\beta_3 = 0.5$ | $\beta_3 = 0.7$ | $\beta_3 = 0.9$ |
|---|---|---|---|---|---|
| $a_n = 2^{n-1}$ | 91.76 | 91.98 | 92.42 | 92.43 | 92.16 |
| $a_n = 3^{n-1}$ | 92.26 | 92.23 | 92.28 | 92.34 | 92.38 |
| $a_n = 4^{n-1}$ | 91.97 | **92.44** | 92.25 | 92.12 | 92.31 |

# D EMPIRICAL EXPERIMENTS

To better understand how different optimizers behave in complex landscapes, we visualize their trajectories on two classical benchmark functions: Rosenbrock and Rastrigin. These functions are used to evaluate the optimizer's ability to escape saddle points, navigate flat valleys, and avoid local minima. Rosenbrock tests the optimizer's capacity to follow narrow curved paths toward a global minimum, while Rastrigin challenges it with a rugged landscape filled with deceptive local minima.

---

[4]https://github.com/openai/improved-diffusion
[5]https://github.com/Liuhong99/Sophia
[6]https://github.com/karpathy/nanoGPT

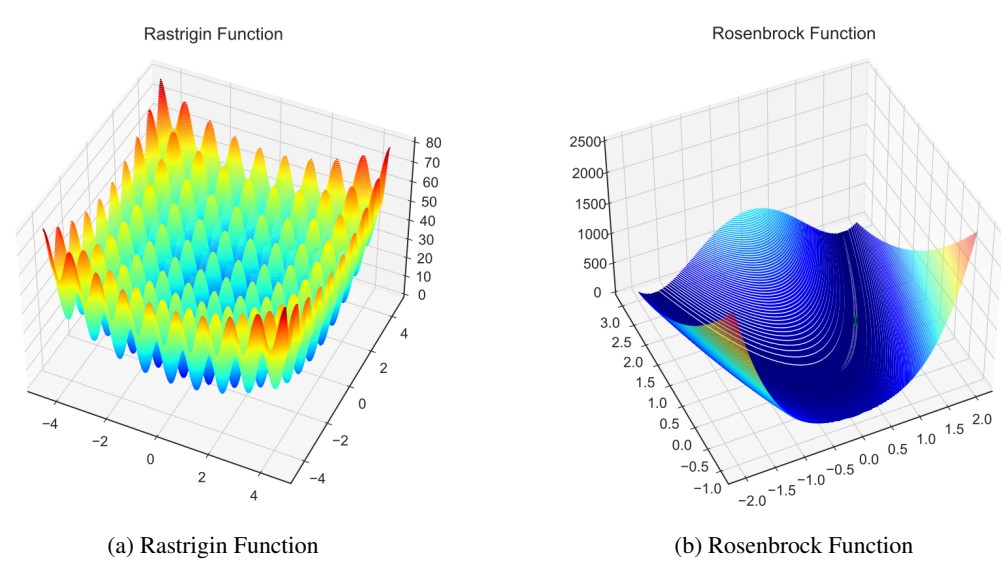

(a) Rastrigin Function          (b) Rosenbrock Function

Figure 5: 3D visualization of benchmark functions

Rastrigin: A highly non-convex function with many local minima. The global minimum is at $(0, 0)$.

Rosenbrock: A narrow, curved valley with the global minimum at $(1, 1)$. It's commonly used to evaluate optimizer stability and curvature sensitivity.

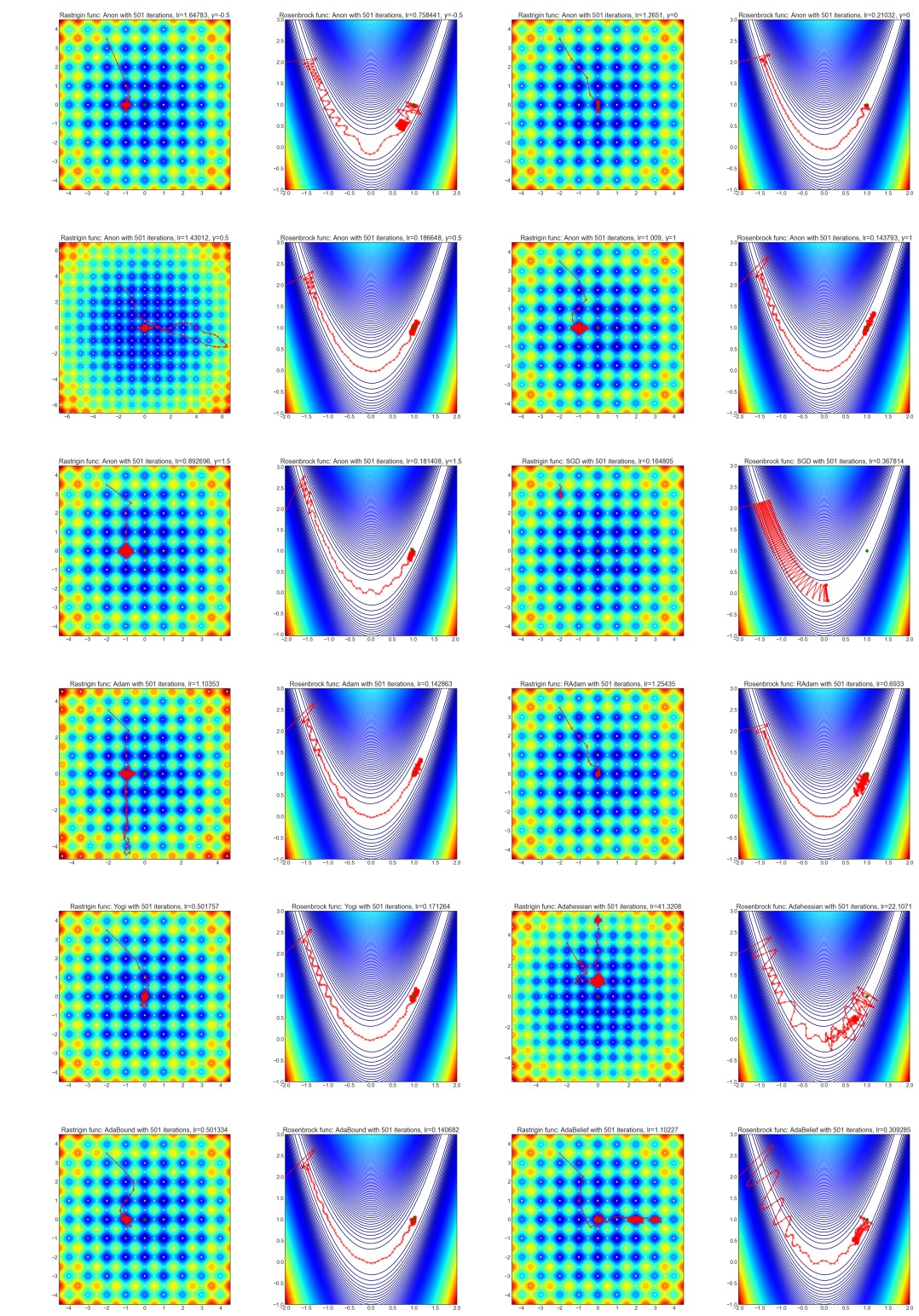

Figure 6: Optimization trajectories comparison under different hyperparameters (searched for each optimizers). The first 10 figures show the optimization trajectories of Anon under different $\gamma$ selections, while the remaining 14 figures display the trajectories of other optimizers.

## E   THEOREM 1 IN MAIN PAPER

**Theorem 5.** *If $\psi$ and $\psi'$ are from the same equivalence class, there is a function $f : \mathbb{N}_+ \to \mathbb{R}_+$ that makes $\psi_n(\boldsymbol{x}_{1:n}) = \psi'_n(\boldsymbol{x}_{1:n})f(n)$ for any $\boldsymbol{x}_{1:n} \in \mathbb{R}^n$ and any $n \in \mathbb{N}_+$.*

*Proof.* Let $h(k; \boldsymbol{g}_{1:n}) = \ln \psi_n(k\boldsymbol{g}_{1:n}) - \ln \psi'_n(k\boldsymbol{g}_{1:n}), h : \mathbb{R} \to \mathbb{R}$. Because $\psi_n$ and $\psi'_n$ are continuous, $h$ is continuous.

When $k \neq 0$, we have

$$
\begin{aligned}
h'(k; \boldsymbol{x}_{1:n}) &= \lim_{\Delta k \to 0} \frac{\ln \psi_n((k + \Delta k)\boldsymbol{x}_{1:n}) - \ln \psi'_n((k + \Delta k)\boldsymbol{x}_{1:n}) - \ln \psi_n(k\boldsymbol{x}_{1:n}) + \ln \psi'_n(k\boldsymbol{x}_{1:n})}{\Delta k} \\
&= \frac{1}{k} \lim_{\Delta k \to 0} \frac{\ln \psi_n((1 + \Delta k/k)k\boldsymbol{x}_{1:n}) - \ln \psi'_n((1 + \Delta k/k)k\boldsymbol{x}_{1:n}) - \ln \psi_n(k\boldsymbol{x}_{1:n}) + \ln \psi'_n(k\boldsymbol{x}_{1:n})}{\Delta k/k} \\
&= \frac{1}{k} \lim_{\Delta k \to 0} \frac{[\ln \psi_n((1 + \Delta k/k)k\boldsymbol{x}_{1:n}) - \ln \psi_n(k\boldsymbol{x}_{1:n})] - [\ln \psi'_n((1 + \Delta k/k)k\boldsymbol{x}_{1:n}) - \ln \psi'_n(k\boldsymbol{x}_{1:n})]}{\Delta k/k} \\
&= \frac{1}{k}\Big[ A_n(\psi, \boldsymbol{x}_{1:n}) - A_n(\psi', \boldsymbol{x}_{1:n}) \Big] \\
&= \frac{1}{k} \cdot 0 \quad \Big( Since\ \psi\ and\ \psi'\ are\ in\ the\ same\ class \Big) \\
&= 0
\end{aligned}
\tag{12}
$$

So $h(k; \boldsymbol{x}_{1:n}) = C_1$ when $k > 0$, $h(k; \boldsymbol{x}_{1:n}) = C_2$ when $k < 0$. And because $h$ is continuous, we have $C_1 = C_2 = h(0; \boldsymbol{x}_{1:n}) = \ln \frac{\psi_n(0)}{\psi'_n(0)}$.

Therefore, we have $\frac{\psi_n(k\boldsymbol{x}_{1:n})}{\psi'_n(k\boldsymbol{x}_{1:n})} = \frac{\psi_n(0)}{\psi'_n(0)}$ for $\forall k \in \mathbb{R}$.

And since $x_{0\,1:n}$ can be any vector $\in \mathbb{R}^n$ and any $n \in \mathbb{N}_+$, we have $\frac{\psi_n(\boldsymbol{x}_{1:n})}{\psi'_n(\boldsymbol{x}_{1:n})} = \frac{\psi_n(0)}{\psi'_n(0)}$ for $\forall \boldsymbol{x}_{1:n} \in \mathbb{R}^n$, $\forall n \in \mathbb{N}_+$.

Let $f(n) = \frac{\psi_n(0)}{\psi'_n(0)}$, we have $\psi_n(\boldsymbol{x}_{1:n}) = \psi'_n(\boldsymbol{x}_{1:n})f(n)$ for any $\boldsymbol{x}_{1:n} \in \mathbb{R}^n$ and any $n \in \mathbb{N}_+$. $\qquad\square$

## F   THEOREM 2 IN MAIN PAPER

**Theorem 6.** *For the optimizer Anon described in Algorithm 2, the adaptivity of Anon in $i$-th dimension is $\in [\gamma(1 - k), \gamma)$, where $k = \epsilon / \min_{j \in [\tilde{a}_t]} EMA(\boldsymbol{g}^2_{a_{j-1}+1:a_j,i}; \beta_2)$.*

*Proof.* We let $f_{n,\gamma}(\boldsymbol{x}) = \beta_3^{-n}(1 - \beta_3 \mathbf{1}_{n>1})EMA^\gamma(\boldsymbol{x}^2_{a_{n-1}+1:a_n} + \epsilon; \beta_2)$, so we have

$$
\begin{aligned}
A(\psi, \boldsymbol{g}_{1:t,i}) &= \nabla_k \ln \left( \sum_{j=1}^{\tilde{a}_t} \beta_3^{\tilde{a}_t} f_{j,\gamma}(k\boldsymbol{g}_{1:t,i}) \right)^{1/2} \Bigg|_{k=1} \\
&= \frac{\gamma \sum_{j=1}^{\tilde{a}_t} \beta_3^{\tilde{a}_t} f_{j,\gamma-1}(\boldsymbol{g}_{1:t,i}) EMA(\boldsymbol{g}^2_{a_{j-1}+1:a_j,i}; \beta_2)}{\sum_{j=1}^{\tilde{a}_t} \beta_3^{\tilde{a}_t} f_{j,\gamma}(\boldsymbol{g}_{1:t,i})} \\
&= \frac{\gamma \sum_{j=1}^{\tilde{a}_t} \beta_3^{\tilde{a}_t} f_{j,\gamma-1}(\boldsymbol{g}_{1:t,i})[EMA(\boldsymbol{g}^2_{a_{j-1}+1:a_j,i} + \epsilon; \beta_2) - \epsilon]}{\sum_{j=1}^{\tilde{a}_t} \beta_3^{\tilde{a}_t} f_{j,\gamma}(\boldsymbol{g}_{1:t,i})} \\
&= \frac{\gamma \sum_{j=1}^{\tilde{a}_t} \beta_3^{\tilde{a}_t} f_{j,\gamma}(\boldsymbol{g}_{1:t,i}) - \gamma\epsilon \sum_{j=1}^{\tilde{a}_t} \beta_3^{\tilde{a}_t} f_{j,\gamma-1}(\boldsymbol{g}_{1:t,i})}{\sum_{j=1}^{\tilde{a}_t} \beta_3^{\tilde{a}_t} f_{j,\gamma}(\boldsymbol{g}_{1:t,i})} \\
&= \gamma \left( 1 - \epsilon \cdot \frac{\sum_{j=1}^{\tilde{a}_t} \beta_3^{\tilde{a}_t} f_{j,\gamma-1}(\boldsymbol{g}_{1:t,i})}{\sum_{j=1}^{\tilde{a}_t} \beta_3^{\tilde{a}_t} f_{j,\gamma}(\boldsymbol{g}_{1:t,i})} \right) \\
&= \gamma \left( 1 - \epsilon \cdot \frac{\sum_{j=1}^{\tilde{a}_t} f_{j,\gamma-1}(\boldsymbol{g}_{1:t,i})}{\sum_{j=1}^{\tilde{a}_t} f_{j,\gamma}(\boldsymbol{g}_{1:t,i})} \right)
\end{aligned}
\tag{13}
$$

$$\geq \gamma(1-k) \quad \left(Since\ k = \epsilon/\min_{j\in[\tilde{a}_t]} EMA(\boldsymbol{g}^2_{a_{j-1}+1:a_j,i}; \beta_2)\right) \tag{14}$$

□

## G  THEOREM 3 IN MAIN PAPER

For simplicity, we omit the debiasing step in theoretical analysis as in Reddi et al. (2019). It is easy to prove that the analysis also applys to the de-biased version.

**Lemma 7.** *(McMahan & Streeter, 2010) For any $Q \in S^d_+$ and convex feasible set $\mathcal{F} \subset \mathbb{R}^d$, suppose $u_1 = \min_{x\in\mathcal{F}} \left\|Q^{1/2}(x-z_1)\right\|$ and $u_2 = \min_{x\in\mathcal{F}} \left\|Q^{1/2}(x-z_2)\right\|$, then we have $\left\|Q^{1/2}(u_1-u_2)\right\| \leq \left\|Q^{1/2}(z_1 - z_2)\right\|$.*

**Theorem 8.** *(Convergence analysis for online convex optimization) Let $\{\theta_t\}$ and $\{v_k\}$ be the sequence obtained by Algorithm 2, $\gamma \in \mathbb{R}$, $\beta_1 \in [0,1)$, $\beta_2 \in [0,1)$, $\beta_{1,t+1} \in [0,\beta_{1,t}]$, $\beta_{1,1} = \beta_1$, $\eta(t) = \frac{\eta_0}{\sqrt{t}}$, for $\forall t \in [T]$. Assume that $\|x-y\|_\infty \leq D_\infty$ for $\forall x, y \in \mathcal{F}$. Suppose $f(\theta)$ is a convex function, $\|g_t\|_\infty \leq G_\infty$, for $\forall t \in [T]$, $\theta \in \mathcal{F}$. Let $C_l = \min(G^{-\gamma}_\infty, \epsilon^{-\gamma})$, $C_u = \max(G^{-\gamma}_\infty, \epsilon^{-\gamma})$, where $\epsilon \in \mathbb{R}_+$ is a very number set in Algorithm 2. The optimal point of $f$ is denoted as $\theta^*$. For $\{\theta_t\}$ generated by Anon, there is a bound on the regret:*

$$\sum_{t=1}^{T}[f_t(\theta_t) - f_t(\theta^*)] \leq \frac{(1-2\sqrt{2})D^2_\infty}{(1-\sqrt{2})(1-\beta_1)C_l\eta_0}\sqrt{T} + \sum_{t=1}^{T-1}\left[\frac{\beta_{1,t+1}\mathbb{I}_{\beta_{1,t+1}>\beta_{1,t}}D^2_\infty}{2C_l\eta_{t+1}(1-\beta_1)^2}\right]$$

$$+ \frac{D^2_\infty}{2C_l\eta_1(1-\beta_1)} + \frac{dD_\infty G_\infty}{1-\beta_1}\sum_{t=1}^{T}\beta_{1,t} + \frac{dG^2_\infty C_u \eta_0}{1-\beta_1}\sqrt{T}$$

*Proof.*

$$\boldsymbol{v}_k = \sqrt{2/(\frac{1}{\boldsymbol{v}^2_{k-1}} + \boldsymbol{\sigma}^\gamma_k)} \text{ if } k > 0 \text{ else } \boldsymbol{\sigma}^{-\gamma/2}_k$$

$$\frac{1}{\boldsymbol{v}^2_k} = \frac{\frac{1}{\boldsymbol{v}^2_{k-1}} + \boldsymbol{\sigma}^\gamma_k}{2} \text{ if } k > 0 \text{ else } \boldsymbol{\sigma}^\gamma_k$$

$$\frac{1}{\boldsymbol{v}^2_k} = \sum_{i=0}^{k} \frac{\boldsymbol{\sigma}^\gamma_i}{2^{\min(k-i+1,k)}}$$

$$\frac{1}{\boldsymbol{v}^2_k} = \sum_{i=0}^{k} \frac{EMA^\gamma(\boldsymbol{g}^2_{\lfloor 2^{k-1}+1\rfloor:2^k} + \epsilon; \beta_2)}{2^{\min(k-i+1,k)}} \tag{15}$$

Since $\|g_t\|_\infty \leq G_\infty$, $C_l = \min(G^{-\gamma}_\infty, \epsilon^{-\gamma})$ and $C_u = \max(G^{-\gamma}_\infty, \epsilon^{-\gamma})$, from 15, we have:

$$\frac{1}{v^2_{k,i}} \in \left[\sum_{i=0}^{k}\frac{C^{-2}_u}{2^{\min(k-i+1,k)}}, \sum_{i=0}^{k}\frac{C^{-2}_l}{2^{\min(k-i+1,k)}}\right]$$

$$\frac{1}{v^2_{k,i}} \in \left[C^{-2}_u, C^{-2}_l\right]$$

$$v_{k,i} \in [C_l, C_u] \tag{16}$$

Let $\eta_t = \eta(t)$.

$$\theta_{t+1} = \prod_{\mathcal{F}, V^{-1}_{\tilde{a}_t}} (\theta_t - \eta_t V_{\tilde{a}_t} m_t) = \min_{\theta\in\mathcal{F}}\left\|V^{-1/2}_{\tilde{a}_t}(\theta - (\theta_t - \eta_t V_{\tilde{a}_t} m_t))\right\|$$

Note that $\prod_{\mathcal{F}, V^{-1}_{\tilde{a}_t}}(\theta^*) = \theta^*$ since $\theta^* \in \mathcal{F}$. Use $\theta^*_i$ and $\theta_{t,i}$ to denote the i-th dimension of $\theta^*$ and $\theta_t$ respectively. From lemma equation 7, using $u_1 = \theta_{t+1}$ and $u_2 = \theta^*$, we have:

$$\left\|V_{\tilde{a}_t}^{-1/2}(\theta_{t+1} - \theta^*)\right\|^2 \leq \left\|V_{\tilde{a}_t}^{-1/2}(\theta_t - \eta_t V_{\tilde{a}_t} m_t - \theta^*)\right\|^2$$

$$= \left\|V_{\tilde{a}_t}^{-1/2}(\theta_t - \theta^*)\right\|^2 + \eta_t^2 \left\|V_{\tilde{a}_t}^{1/2} m_t\right\|^2 - 2\eta_t \langle m_t, \theta_t - \theta^* \rangle$$

$$= \left\|V_{\tilde{a}_t}^{-1/2}(\theta_t - \theta^*)\right\|^2 + \eta_t^2 \left\|V_{\tilde{a}_t}^{1/2} m_t\right\|^2$$

$$- 2\eta_t \langle \beta_{1,t} m_{t-1} + (1 - \beta_{1,t}) g_t, \theta_t - \theta^* \rangle \tag{17}$$

Note that $\beta_1 \in [0, 1)$ and $\beta_2 \in [0, 1)$, rearranging inequality equation 17, we have:

$$\langle g_t, \theta_t - \theta^* \rangle \leq \frac{1}{2\eta_t(1 - \beta_{1,t})}\left(\left\|V_{\tilde{a}_t}^{-1/2}(\theta_t - \theta^*)\right\|^2 - \left\|V_{\tilde{a}_t}^{-1/2}(\theta_{t+1} - \theta^*)\right\|^2\right)$$

$$+ \frac{\eta_t}{2(1 - \beta_{1,t})}\left\|V_{\tilde{a}_t}^{1/2} m_t\right\|^2 + \frac{\beta_{1,t}}{1 - \beta_{1,t}}\langle m_{t-1}, \theta^* - \theta_t \rangle$$

$$\leq \frac{1}{2\eta_t(1 - \beta_{1,t})}\left(\left\|V_{\tilde{a}_t}^{-1/2}(\theta_t - \theta^*)\right\|^2 - \left\|V_{\tilde{a}_t}^{-1/2}(\theta_{t+1} - \theta^*)\right\|^2\right)$$

$$+ \frac{\eta_t}{2(1 - \beta_{1,t})}\left\|V_{\tilde{a}_t}^{1/2} m_t\right\|^2 + \frac{\beta_{1,t}}{1 - \beta_{1,t}}\left\|m_{t-1}\right\|\left\|\theta^* - \theta_t\right\|$$

$$\left(\text{Cauchy-Schwartz's inequality: } \langle u, v \rangle \leq \left\|u\right\|\left\|v\right\|\right)$$

$$\leq \frac{1}{2\eta_t(1 - \beta_{1,t})}\left(\left\|V_{\tilde{a}_t}^{-1/2}(\theta_t - \theta^*)\right\|^2 - \left\|V_{\tilde{a}_t}^{-1/2}(\theta_{t+1} - \theta^*)\right\|^2\right)$$

$$+ \frac{\eta_t}{2(1 - \beta_{1,t})}\left\|V_{\tilde{a}_t}^{1/2} m_t\right\|^2 + \frac{\beta_{1,t}}{1 - \beta_{1,t}}\left\|m_{t-1}\right\|\sqrt{d}D_\infty$$

$$\left(\text{Since } \|x - y\|_\infty \leq D_\infty, \text{ for } \forall x, y \in \mathcal{F}\right)$$

$$= \frac{1}{2\eta_t(1 - \beta_{1,t})}\left(\left\|V_{\tilde{a}_t}^{-1/2}(\theta_t - \theta^*)\right\|^2 - \left\|V_{\tilde{a}_t}^{-1/2}(\theta_{t+1} - \theta^*)\right\|^2\right)$$

$$+ \frac{\eta_t}{2(1 - \beta_{1,t})}\left\|V_{\tilde{a}_t}^{1/2} m_t\right\|^2 + \frac{\beta_{1,t}\sqrt{d}D_\infty}{1 - \beta_{1,t}}\sqrt{\sum_{i=1}^{d} EMA^2(g_{1:t-1,i}; \beta_2)}$$

$$\leq \frac{1}{2\eta_t(1 - \beta_{1,t})}\left(\left\|V_{\tilde{a}_t}^{-1/2}(\theta_t - \theta^*)\right\|^2 - \left\|V_{\tilde{a}_t}^{-1/2}(\theta_{t+1} - \theta^*)\right\|^2\right)$$

$$+ \frac{\eta_t}{2(1 - \beta_{1,t})}\left\|V_{\tilde{a}_t}^{1/2} m_t\right\|^2 + \frac{\beta_{1,t}\sqrt{d}D_\infty}{1 - \beta_{1,t}}\sqrt{\sum_{i=1}^{d} G_\infty^2}$$

$$\left(\text{Since } \|g_t\|_\infty \leq G_\infty\right)$$

$$\leq \frac{1}{2\eta_t(1 - \beta_{1,t})}\left(\left\|V_{\tilde{a}_t}^{-1/2}(\theta_t - \theta^*)\right\|^2 - \left\|V_{\tilde{a}_t}^{-1/2}(\theta_{t+1} - \theta^*)\right\|^2\right)$$

$$+ \frac{\eta_t}{2(1 - \beta_{1,t})}\left\|V_{\tilde{a}_t}^{1/2} m_t\right\|^2 + \frac{\beta_{1,t}dD_\infty}{1 - \beta_{1,t}}G_\infty$$

$$= \frac{1}{2\eta_t(1 - \beta_{1,t})}\left(\left\|V_{\tilde{a}_t}^{-1/2}(\theta_t - \theta^*)\right\|^2 - \left\|V_{\tilde{a}_t}^{-1/2}(\theta_{t+1} - \theta^*)\right\|^2\right)$$

$$+ \frac{\beta_{1,t}dD_\infty G_\infty}{1 - \beta_{1,t}} + \frac{\eta_t}{2(1 - \beta_{1,t})}m_t^\top V_{\tilde{a}_t} m_t$$

$$= \frac{1}{2\eta_t(1 - \beta_{1,t})}\left(\left\|V_{\tilde{a}_t}^{-1/2}(\theta_t - \theta^*)\right\|^2 - \left\|V_{\tilde{a}_t}^{-1/2}(\theta_{t+1} - \theta^*)\right\|^2\right)$$

$$+ \frac{\beta_{1,t}dD_\infty G_\infty}{1 - \beta_{1,t}} + \frac{\eta_t}{2(1 - \beta_{1,t})}\sum_{i=1}^{d} m_{t,i}^2 v_{\tilde{a}_t,i}$$

$$\leq \frac{1}{2\eta_t(1-\beta_{1,t})}\left(\left\|V_{\tilde{a}_t}^{-1/2}(\theta_t-\theta^*)\right\|^2 - \left\|V_{\tilde{a}_t}^{-1/2}(\theta_{t+1}-\theta^*)\right\|^2\right)$$

$$+ \frac{\beta_{1,t}dD_\infty G_\infty}{1-\beta_{1,t}} + \frac{\eta_t}{2(1-\beta_{1,t})}\sum_{i=1}^d m_{t,i}^2 C_u$$

$$\left(Apply\,formula\,equation\,16\right)$$

$$\leq \frac{1}{2\eta_t(1-\beta_{1,t})}\left(\left\|V_{\tilde{a}_t}^{-1/2}(\theta_t-\theta^*)\right\|^2 - \left\|V_{\tilde{a}_t}^{-1/2}(\theta_{t+1}-\theta^*)\right\|^2\right)$$

$$+ \frac{\beta_{1,t}dD_\infty G_\infty}{1-\beta_{1,t}} + \frac{dG_\infty^2 C_u \eta_t}{2(1-\beta_{1,t})} \tag{18}$$

By convexity of $f$, we have:

$$\sum_{t=1}^T f_t(\theta_t) - f_t(\theta^*) \leq \sum_{t=1}^T \langle g_t, \theta_t - \theta^*\rangle$$

$$\leq \sum_{t=1}^T \left[\frac{1}{2\eta_t(1-\beta_{1,t})}\left(\left\|V_{\tilde{a}_t}^{-1/2}(\theta_t-\theta^*)\right\|^2 - \left\|V_{\tilde{a}_t}^{-1/2}(\theta_{t+1}-\theta^*)\right\|^2\right)\right.$$

$$\left.+ \frac{\beta_{1,t}dD_\infty G_\infty}{1-\beta_{1,t}} + \frac{dG_\infty^2 C_u \eta_t}{2(1-\beta_{1,t})}\right]$$

$$\left(By\,formula\,equation\,18\right)$$

$$\leq \sum_{t=1}^T \left[\frac{1}{2\eta_t(1-\beta_{1,t})}\left(\left\|V_{\tilde{a}_t}^{-1/2}(\theta_t-\theta^*)\right\|^2 - \left\|V_{\tilde{a}_t}^{-1/2}(\theta_{t+1}-\theta^*)\right\|^2\right)\right]$$

$$+ \frac{1}{1-\beta_1}\sum_{t=1}^T \left(\beta_{1,t}dD_\infty G_\infty + \frac{dG_\infty^2 C_u \eta_t}{2}\right)$$

$$\left(Since\,0\leq\beta_{1,t}\leq\beta_1<1\right)$$

$$= \sum_{t=1}^T \left[\frac{1}{2\eta_t(1-\beta_{1,t})}\left(\left\|V_{\tilde{a}_t}^{-1/2}(\theta_t-\theta^*)\right\|^2 - \left\|V_{\tilde{a}_t}^{-1/2}(\theta_{t+1}-\theta^*)\right\|^2\right)\right]$$

$$+ \frac{1}{1-\beta_1}\sum_{t=1}^T \left(\beta_{1,t}dD_\infty G_\infty + \frac{dG_\infty^2 C_u \eta_0}{2\sqrt{t}}\right)$$

$$\leq \sum_{t=1}^T \left[\frac{1}{2\eta_t(1-\beta_{1,t})}\left(\left\|V_{\tilde{a}_t}^{-1/2}(\theta_t-\theta^*)\right\|^2 - \left\|V_{\tilde{a}_t}^{-1/2}(\theta_{t+1}-\theta^*)\right\|^2\right)\right]$$

$$+ \frac{dD_\infty G_\infty}{1-\beta_1}\sum_{t=1}^T \beta_{1,t} + \frac{dG_\infty^2 C_u \eta_0}{1-\beta_1}\int_0^T \frac{1}{2\sqrt{t}}\,\mathrm{d}t$$

$$\left(Since\,\eta_t = \eta_0/\sqrt{t}\right)$$

$$= \sum_{t=1}^T \left[\frac{1}{2\eta_t(1-\beta_{1,t})}\left(\left\|V_{\tilde{a}_t}^{-1/2}(\theta_t-\theta^*)\right\|^2 - \left\|V_{\tilde{a}_t}^{-1/2}(\theta_{t+1}-\theta^*)\right\|^2\right)\right]$$

$$+ \frac{dD_\infty G_\infty}{1-\beta_1}\sum_{t=1}^T \beta_{1,t} + \frac{dG_\infty^2 C_u \eta_0}{1-\beta_1}\sqrt{T}$$

$$\leq \sum_{t=1}^{T-1} \left[\frac{1}{2\eta_{t+1}(1-\beta_{1,t+1})}\left\|V_{\tilde{a}_{t+1}}^{-1/2}(\theta_{t+1}-\theta^*)\right\|^2 - \frac{1}{2\eta_t(1-\beta_{1,t})}\left\|V_{\tilde{a}_t}^{-1/2}(\theta_{t+1}-\theta^*)\right\|^2\right]$$

$$+ \frac{1}{2\eta_1(1-\beta_1)}\left\|V_1^{-1/2}(\theta_1-\theta^*)\right\|^2 + \frac{dD_\infty G_\infty}{1-\beta_1}\sum_{t=1}^{T}\beta_{1,t} + \frac{dG_\infty^2 C_u \eta_0}{1-\beta_1}\sqrt{T}$$

$$= \sum_{t=1}^{T-1}\left[\frac{1}{2\eta_{t+1}(1-\beta_{1,t})}\left\|V_{\tilde{a}_{t+1}}^{-1/2}(\theta_{t+1}-\theta^*)\right\|^2 - \frac{1}{2\eta_t(1-\beta_{1,t})}\left\|V_{\tilde{a}_t}^{-1/2}(\theta_{t+1}-\theta^*)\right\|^2\right.$$

$$\left.+ \frac{\beta_{1,t+1}-\beta_{1,t}}{2\eta_{t+1}(1-\beta_{1,t})(1-\beta_{1,t+1})}\left\|V_{\tilde{a}_{t+1}}^{-1/2}(\theta_{t+1}-\theta^*)\right\|^2\right]$$

$$+ \frac{1}{2\eta_1(1-\beta_1)}\left\|V_1^{-1/2}(\theta_1-\theta^*)\right\|^2 + \frac{dD_\infty G_\infty}{1-\beta_1}\sum_{t=1}^{T}\beta_{1,t} + \frac{dG_\infty^2 C_u \eta_0}{1-\beta_1}\sqrt{T}$$

$$= \sum_{t=1}^{T-1}\left\{\frac{1}{2(1-\beta_{1,t})}\left[(\theta_{t+1}-\theta^*)^\top\left(\frac{V_{\tilde{a}_{t+1}}^{-1}}{\eta_{t+1}} - \frac{V_{\tilde{a}_t}^{-1}}{\eta_t}\right)(\theta_{t+1}-\theta^*)\right]\right\}$$

$$+ \sum_{t=1}^{T-1}\left[\frac{\beta_{1,t+1}-\beta_{1,t}}{2\eta_{t+1}(1-\beta_{1,t})(1-\beta_{1,t+1})}\left\|V_{\tilde{a}_{t+1}}^{-1/2}(\theta_{t+1}-\theta^*)\right\|^2\right]$$

$$+ \frac{1}{2\eta_1(1-\beta_{1,t})}\left\|V_1^{-1/2}(\theta_1-\theta^*)\right\|^2 + \frac{dD_\infty G_\infty}{1-\beta_1}\sum_{t=1}^{T}\beta_{1,t} + \frac{dG_\infty^2 C_u \eta_0}{1-\beta_1}\sqrt{T}$$

$$= \sum_{k=1}^{\tilde{a}_T}\sum_{t=a_k}^{\min(T,a_{k+1})-1}\left\{\frac{1}{2(1-\beta_{1,t})}\left[(\theta_{t+1}-\theta^*)^\top\left(\frac{V_{\tilde{a}_{t+1}}^{-1}}{\eta_{t+1}} - \frac{V_{\tilde{a}_t}^{-1}}{\eta_t}\right)(\theta_{t+1}-\theta^*)\right]\right\}$$

$$+ \sum_{t=1}^{T-1}\left[\frac{\beta_{1,t+1}-\beta_{1,t}}{2\eta_{t+1}(1-\beta_{1,t})(1-\beta_{1,t+1})}\left\|V_{\tilde{a}_{t+1}}^{-1/2}(\theta_{t+1}-\theta^*)\right\|^2\right]$$

$$+ \frac{1}{2\eta_1(1-\beta_1)}\left\|V_1^{-1/2}(\theta_1-\theta^*)\right\|^2 + \frac{dD_\infty G_\infty}{1-\beta_1}\sum_{t=1}^{T}\beta_{1,t} + \frac{dG_\infty^2 C_u \eta_0}{1-\beta_1}\sqrt{T}$$

$$= \sum_{k=1}^{\tilde{a}_T}\sum_{t=a_k}^{\min(T,a_{k+1})-2}\left\{\frac{1}{2(1-\beta_{1,t})}\left[(\theta_{t+1}-\theta^*)^\top\left(\frac{V_{\tilde{a}_{t+1}}^{-1}}{\eta_{t+1}} - \frac{V_{\tilde{a}_t}^{-1}}{\eta_t}\right)(\theta_{t+1}-\theta^*)\right]\right\}$$

$$+ \sum_{k=1}^{\tilde{a}_T-1}\left\{\frac{1}{2(1-\beta_{1a_{k+1}-1})}\left[(\theta_{a_{k+1}}-\theta^*)^\top\left(\frac{V_{k+1}^{-1}}{\eta_{a_{k+1}}} - \frac{V_k^{-1}}{\eta_{a_{k+1}-1}}\right)(\theta_{a_{k+1}}-\theta^*)\right]\right\}$$

$$+ \sum_{t=1}^{T-1}\left[\frac{\beta_{1,t+1}-\beta_{1,t}}{2\eta_{t+1}(1-\beta_{1,t})(1-\beta_{1,t+1})}\left\|V_{\tilde{a}_{t+1}}^{-1/2}(\theta_{t+1}-\theta^*)\right\|^2\right]$$

$$+ \frac{1}{2\eta_1(1-\beta_1)}\left\|V_1^{-1/2}(\theta_1-\theta^*)\right\|^2 + \frac{dD_\infty G_\infty}{1-\beta_1}\sum_{t=1}^{T}\beta_{1,t} + \frac{dG_\infty^2 C_u \eta_0}{1-\beta_1}\sqrt{T}$$

$$= \sum_{k=1}^{\tilde{a}_T}\sum_{t=a_k}^{\min(T,a_{k+1})-2}\left\{\frac{1}{2(1-\beta_{1,t})}\left[(\theta_{t+1}-\theta^*)^\top\left(\frac{V_k^{-1}}{\eta_{t+1}} - \frac{V_k^{-1}}{\eta_t}\right)(\theta_{t+1}-\theta^*)\right]\right\}$$

$$+ \sum_{k=1}^{\tilde{a}_T-1}\left\{\frac{1}{2(1-\beta_{1a_{k+1}-1})}\left[(\theta_{a_{k+1}}-\theta^*)^\top\left(\frac{V_{k+1}^{-1}}{\eta_{a_{k+1}}} - \frac{V_k^{-1}}{\eta_{a_{k+1}-1}}\right)(\theta_{a_{k+1}}-\theta^*)\right]\right\}$$

$$+ \sum_{t=1}^{T-1}\left[\frac{\beta_{1,t+1}-\beta_{1,t}}{2\eta_{t+1}(1-\beta_{1,t})(1-\beta_{1,t+1})}\left\|V_{\tilde{a}_{t+1}}^{-1/2}(\theta_{t+1}-\theta^*)\right\|^2\right]$$

$$+ \frac{1}{2\eta_1(1-\beta_1)}\left\|V_1^{-1/2}(\theta_1-\theta^*)\right\|^2 + \frac{dD_\infty G_\infty}{1-\beta_1}\sum_{t=1}^{T}\beta_{1,t} + \frac{dG_\infty^2 C_u \eta_0}{1-\beta_1}\sqrt{T}$$

$$\leq \sum_{k=1}^{\tilde{a}_T} \sum_{t=a_k}^{\min(T, a_{k+1})-2} \left\{ \frac{1}{2(1-\beta_1)} \left[ D_\infty e_d^\top \left( \frac{V_k^{-1}}{\eta_{t+1}} - \frac{V_k^{-1}}{\eta_t} \right) D_\infty e_d \right] \right\}$$

$$+ \sum_{k=1}^{\tilde{a}_T-1} \left\{ \frac{1}{2(1-\beta_1)} \left[ D_\infty e_d^\top \left( \frac{C_l^{-1} \boldsymbol{I}_d}{\eta_{a_{k+1}}} \right) D_\infty e_d \right] \right\}$$

$$+ \sum_{t=1}^{T-1} \left[ \frac{\beta_{1,t+1} - \beta_{1,t}}{2\eta_{t+1}(1-\beta_{1,t})(1-\beta_{1,t+1})} \left\| V_{\tilde{a}_{t+1}}^{-1/2}(\theta_{t+1} - \theta^*) \right\|^2 \right]$$

$$+ \frac{1}{2\eta_1(1-\beta_1)} \left\| V_1^{-1/2}(\theta_1 - \theta^*) \right\|^2 + \frac{dD_\infty G_\infty}{1-\beta_1} \sum_{t=1}^{T} \beta_{1,t} + \frac{dG_\infty^2 C_u \eta_0}{1-\beta_1} \sqrt{T}$$

$$\left( \textit{Since } \eta_t = \eta_0/\sqrt{t}, \text{ and } 0 \leq \beta_{1,t} \leq \beta_1 < 1 \right)$$

$$= \sum_{k=1}^{\tilde{a}_T-1} \left\{ \frac{1}{2(1-\beta_1)} \left[ D_\infty e_d^\top \left( \frac{V_k^{-1}}{\eta_{a_{k+1}-1}} - \frac{V_k^{-1}}{\eta_{a_k}} \right) D_\infty e_d \right] \right\}$$

$$+ \frac{1}{2(1-\beta_1)} \left[ D_\infty e_d^\top \left( \frac{V_{\tilde{a}_T}^{-1}}{\eta_T} - \frac{V_{\tilde{a}_T}^{-1}}{\eta_{a_{\tilde{a}_T}}} \right) D_\infty e_d \right]$$

$$+ \sum_{k=1}^{\tilde{a}_T-1} \left\{ \frac{1}{2(1-\beta_1)} \left[ D_\infty e_d^\top \left( \frac{C_l^{-1} \boldsymbol{I}_d}{\eta_{a_{k+1}}} \right) D_\infty e_d \right] \right\}$$

$$+ \sum_{t=1}^{T-1} \left[ \frac{\beta_{1,t+1} - \beta_{1,t}}{2\eta_{t+1}(1-\beta_{1,t})(1-\beta_{1,t+1})} \left\| V_{\tilde{a}_{t+1}}^{-1/2}(\theta_{t+1} - \theta^*) \right\|^2 \right]$$

$$+ \frac{1}{2\eta_1(1-\beta_1)} \left\| V_1^{-1/2}(\theta_1 - \theta^*) \right\|^2 + \frac{dD_\infty G_\infty}{1-\beta_1} \sum_{t=1}^{T} \beta_{1,t} + \frac{dG_\infty^2 C_u \eta_0}{1-\beta_1} \sqrt{T}$$

$$\leq 2 \sum_{k=1}^{\tilde{a}_T-1} \left\{ \frac{1}{2(1-\beta_1)} \left[ D_\infty e_d^\top \left( \frac{C_l^{-1} \boldsymbol{I}_d}{\eta_{a_{k+1}}} \right) D_\infty e_d \right] \right\}$$

$$+ \frac{1}{2(1-\beta_1)} \left[ D_\infty e_d^\top \left( \frac{C_l^{-1} \boldsymbol{I}_d}{\eta_T} \right) D_\infty e_d \right]$$

$$+ \sum_{t=1}^{T-1} \left[ \frac{\beta_{1,t+1} - \beta_{1,t}}{2\eta_{t+1}(1-\beta_{1,t})(1-\beta_{1,t+1})} \left\| V_{\tilde{a}_{t+1}}^{-1/2}(\theta_{t+1} - \theta^*) \right\|^2 \right]$$

$$+ \frac{1}{2\eta_1(1-\beta_1)} \left\| V_1^{-1/2}(\theta_1 - \theta^*) \right\|^2 + \frac{dD_\infty G_\infty}{1-\beta_1} \sum_{t=1}^{T} \beta_{1,t} + \frac{dG_\infty^2 C_u \eta_0}{1-\beta_1} \sqrt{T}$$

$$\leq \frac{dD_\infty^2 C_l^{-1}}{(1-\beta_1)\eta_0} \sum_{k=1}^{\tilde{a}_T-1} \sqrt{a_{k+1}}$$

$$+ \frac{dD_\infty^2 C_l^{-1}}{(1-\beta_1)\eta_0} \sqrt{T}$$

$$+ \sum_{t=1}^{T-1} \left[ \frac{\beta_{1,t+1} - \beta_{1,t}}{2\eta_{t+1}(1-\beta_{1,t})(1-\beta_{1,t+1})} \left\| V_{\tilde{a}_{t+1}}^{-1/2}(\theta_{t+1} - \theta^*) \right\|^2 \right]$$

$$+ \frac{1}{2\eta_1(1-\beta_1)} \left\| V_1^{-1/2}(\theta_1 - \theta^*) \right\|^2 + \frac{dD_\infty G_\infty}{1-\beta_1} \sum_{t=1}^{T} \beta_{1,t} + \frac{dG_\infty^2 C_u \eta_0}{1-\beta_1} \sqrt{T}$$

$$\leq \frac{dD_\infty^2 C_l^{-1}}{(1-\beta_1)\eta_0} \left( \sqrt{T} + \sum_{k=1}^{\tilde{a}_T-1} \sqrt{a_{k+1}} \right)$$

$$+ \sum_{t=1}^{T-1} \left[ \frac{\beta_{1,t+1} - \beta_{1,t}}{2\eta_{t+1}(1-\beta_{1,t})(1-\beta_{1,t+1})} \left\| V_{\tilde{a}_{t+1}}^{-1/2}(\theta_{t+1} - \theta^*) \right\|^2 \right]$$

$$+ \frac{1}{2\eta_1(1-\beta_1)} \left\| V_1^{-1/2}(\theta_1 - \theta^*) \right\|^2 + \frac{dD_\infty G_\infty}{1-\beta_1} \sum_{t=1}^{T} \beta_{1,t} + \frac{dG_\infty^2 C_u \eta_0}{1-\beta_1} \sqrt{T}$$

$$\leq \frac{dD_\infty^2 C_l^{-1}}{(1-\beta_1)\eta_0} \left( \sqrt{T} + \sum_{k=1}^{\tilde{a}_T - 1} \sqrt{a_{k+1}} \right) + \sum_{t=1}^{T-1} \left[ \frac{\beta_{1,t+1} - \beta_{1,t}}{2\eta_{t+1}(1-\beta_{1,t})(1-\beta_{1,t+1})} \left\| V_{\tilde{a}_{t+1}}^{-1/2}(\theta_{t+1} - \theta^*) \right\|^2 \right]$$

$$+ \frac{1}{2\eta_1(1-\beta_1)} \left( D_\infty e_d^\top V_1^{-1} D_\infty e_d \right) + \frac{dD_\infty G_\infty}{1-\beta_1} \sum_{t=1}^{T} \beta_{1,t} + \frac{dG_\infty^2 C_u \eta_0}{1-\beta_1} \sqrt{T}$$

$$\leq \frac{dD_\infty^2 C_l^{-1}}{(1-\beta_1)\eta_0} \left( \sqrt{T} + \sum_{k=1}^{\tilde{a}_T - 1} \sqrt{a_{k+1}} \right) + \sum_{t=1}^{T-1} \left[ \frac{\beta_{1,t+1} - \beta_{1,t}}{2\eta_{t+1}(1-\beta_{1,t})(1-\beta_{1,t+1})} \left\| V_{\tilde{a}_{t+1}}^{-1/2}(\theta_{t+1} - \theta^*) \right\|^2 \right]$$

$$+ \frac{dD_\infty^2 C_l^{-1}}{2\eta_1(1-\beta_1)} + \frac{dD_\infty G_\infty}{1-\beta_1} \sum_{t=1}^{T} \beta_{1,t} + \frac{dG_\infty^2 C_u \eta_0}{1-\beta_1} \sqrt{T}$$

$$\leq \frac{dD_\infty^2 C_l^{-1}}{(1-\beta_1)\eta_0} \left( \sqrt{T} + \sum_{k=1}^{\tilde{a}_T - 1} \sqrt{a_{k+1}} \right) + \sum_{t=1}^{T-1} \left[ \frac{\beta_{1,t+1} \mathbb{I}_{\beta_{1,t+1} > \beta_{1,t}}}{2\eta_{t+1}(1-\beta_{1,t})(1-\beta_{1,t+1})} \left\| V_{\tilde{a}_{t+1}}^{-1/2}(\theta_{t+1} - \theta^*) \right\|^2 \right]$$

$$+ \frac{dD_\infty^2 C_l^{-1}}{2\eta_1(1-\beta_1)} + \frac{dD_\infty G_\infty}{1-\beta_1} \sum_{t=1}^{T} \beta_{1,t} + \frac{dG_\infty^2 C_u \eta_0}{1-\beta_1} \sqrt{T}$$

$$\leq \frac{dD_\infty^2 C_l^{-1}}{(1-\beta_1)\eta_0} \left( \sqrt{T} + \sum_{k=1}^{\tilde{a}_T - 1} \sqrt{a_{k+1}} \right) + \sum_{t=1}^{T-1} \left[ \frac{\beta_{1,t+1} \mathbf{1}_{\beta_{1,t+1} > \beta_{1,t}}}{2\eta_{t+1}(1-\beta_1)^2} \left( D_\infty e_d^\top C_l^{-1} I_d D_\infty e_d \right) \right]$$

$$+ \frac{dD_\infty^2 C_l^{-1}}{2\eta_1(1-\beta_1)} + \frac{dD_\infty G_\infty}{1-\beta_1} \sum_{t=1}^{T} \beta_{1,t} + \frac{dG_\infty^2 C_u \eta_0}{1-\beta_1} \sqrt{T}$$

$$\leq \frac{dD_\infty^2 C_l^{-1}}{(1-\beta_1)\eta_0} \left( \sqrt{T} + \sum_{k=1}^{\tilde{a}_T - 1} \sqrt{a_{k+1}} \right) + \sum_{t=1}^{T-1} \left[ \frac{\beta_{1,t+1} \mathbf{1}_{\beta_{1,t+1} > \beta_{1,t}} dD_\infty^2}{2C_l \eta_{t+1}(1-\beta_1)^2} \right]$$

$$+ \frac{dD_\infty^2}{2C_l \eta_1(1-\beta_1)} + \frac{dD_\infty G_\infty}{1-\beta_1} \sum_{t=1}^{T} \beta_{1,t} + \frac{dG_\infty^2 C_u \eta_0}{1-\beta_1} \sqrt{T} \qquad (19)$$

$\square$

**Corollary 8.1.** *Suppose $\beta_{1,t} = \beta_1 \lambda^t,\ \ 0 < \lambda < 1$ in Theorem 8, then we have:*

$$\sum_{t=1}^{T} f_t(\theta_t) - f_t(\theta^*) \leq \frac{dD_\infty^2 C_l^{-1}}{(1-\beta_1)\eta_0} \left( \sqrt{T} + \sum_{k=1}^{\tilde{a}_T - 1} \sqrt{a_{k+1}} \right) + \frac{dD_\infty^2}{2C_l \eta_1(1-\beta_1)}$$

$$+ \frac{dD_\infty G_\infty \beta_1}{(1-\beta_1)(1-\lambda)} + \frac{dG_\infty^2 C_u \eta_0}{1-\beta_1} \sqrt{T} \qquad (20)$$

*Proof.* It is easy to prove using:

$$\sum_{t=1}^{T} \beta_{1,t} = \sum_{t=1}^{T} \beta_1 \lambda^{t-1} < \sum_{t=1}^{\infty} \beta_1 \lambda^{t-1} \leq \frac{\beta_1}{1-\lambda} \qquad (21)$$

Plugging equation 21 into equation 19, we can derive the results above. $\square$

**Corollary 8.2.** *Suppose $a_n = 2^{n-1},\ \ \beta_3 = \frac{1}{2}$ in equation 20, then we have:*

$$\sum_{t=1}^{T} f_t(\theta_t) - f_t(\theta^*) \leq \frac{(1 - 2\sqrt{2})D_\infty^2}{(1-\sqrt{2})(1-\beta_1)C_l \eta_0} \sqrt{T} + \frac{D_\infty^2}{2C_l \eta_1(1-\beta_1)}$$

$$+ \frac{dD_\infty G_\infty \beta_1}{(1-\beta_1)(1-\lambda)} + \frac{dG_\infty^2 C_u \eta_0}{1-\beta_1}\sqrt{T} \tag{22}$$

*Proof.* It is easy to prove using:

$$\sum_{t=1}^{T} a^{t-1} = \frac{1-a^T}{1-a} \tag{23}$$

□

## H   THEOREM 4 IN MAIN PAPER

**Lemma 9.** *(Zhuang et al., 2021) Let $m_t = \beta_1 m_{t-1} + (1-\beta_1)g_t$, let $Q_t \in \mathbb{R}^d$, then*

$$\left\langle Q_t, g_t \right\rangle = \frac{1}{1-\beta_1}\left(\left\langle Q_t, m_t \right\rangle - \left\langle Q_{t-1}, m_{t-1} \right\rangle\right) + \left\langle Q_{t-1}, m_{t-1} \right\rangle + \frac{\beta_1}{1-\beta_1}\left\langle Q_{t-1} - Q_t, m_{t-1} \right\rangle \tag{24}$$

**Theorem 10.** *(Convergence analysis for non-convex stochastic optimization) The update of $\theta_t$ can be described as $\theta_{t+1} = \theta_t - \eta_t V_{\tilde{a}_t} m_t$, and $m_t = \beta_1 m_{t-1} + (1-\beta_1)g_t$.*
*Under the assumptions:*

- *$f$ is differentiable and $f^* \le f \le F$. $\nabla f(x)$ is L-Lipschitz continuous, i.e. $\|\nabla f(x) - \nabla f(y)\| \le L\|x-y\|$, $\forall x, y$.*
- *The noisy gradient is unbias and its infinity norm is bounded by N, i.e. $\mathbb{E}g_t = \nabla f(x)$, $\|g_t\|_\infty \le N$.*

*The hyperparameters are set as: $\eta_t = \eta_0 t^{-p}$, $\eta_0 > 0$, $p \in (0,1)$ where the bounds are $C_l I \preceq V_{\tilde{a}_t} \preceq C_u I$, and $0 < C_l < C_u$ ($A \preceq B$ means $B - A$ is a positive semi-definite matrix). And the $\epsilon$ and $N$ ensure $C_l$ and $C_u$ exist. For sequence $\{\theta_t\}$ generated by Anon, we have:*

$$\frac{1}{T}\sum_{t=1}^{T}\left\|\nabla f(x_t)\right\|^2 \le \frac{1}{\eta_0 C_l}T^{p-1}\left(F - f^* + K\int_1^T t^{-2p}\,\mathrm{d}t + J + K\right)$$

*where*

$$J = \frac{\beta_1^2 d}{4L(1-\beta_1)^2}N^2 + \frac{3dN^2}{1-\beta_1}\eta_0 C_u \sum_{k=1}^{\tilde{a}_t}(a_k - \mathbf{1}_{k\neq 1})^{-p}, \quad K = \left(\frac{1}{1-\beta_1} + \frac{1}{2}\right)L\eta_0^2 N^2 C_u^2 d$$

*Proof.* Let $A_t = V_{\tilde{a}_t}$, $Q_t = \eta_t A_t \nabla f(x_t)$ and let $Q_0 = Q_1$, we have

$$\sum_{t=1}^{T}\left\langle Q_t, g_t \right\rangle = \frac{1}{1-\beta_1}\left\langle Q_T, m_T \right\rangle + \sum_{t=1}^{T}\left\langle Q_{t-1}, m_{t-1} \right\rangle + \frac{\beta_1}{1-\beta_1}\sum_{t=1}^{T}\left\langle Q_{t-1} - Q_t, m_{t-1} \right\rangle$$

$$= \frac{\beta_1}{1-\beta_1}\left\langle Q_T, m_T \right\rangle + \sum_{t=1}^{T}\left\langle Q_t, m_t \right\rangle + \frac{\beta_1}{1-\beta_1}\sum_{t=0}^{T-1}\left\langle Q_t - Q_{t+1}, m_t \right\rangle \tag{25}$$

First we derive a lower bound for equation 25.

$$\begin{aligned}
\left\langle Q_t, g_t \right\rangle &= \left\langle \eta_t A_t \nabla f(x_t), g_t \right\rangle \\
&= \left\langle \eta_{t-1} A_{t-1} \nabla f(x_t), g_t \right\rangle - \left\langle (\eta_{t-1}A_{t-1} - \eta_t A_t)\nabla f(x_t), g_t \right\rangle \\
&\ge \left\langle \eta_{t-1} A_{t-1} \nabla f(x_t), g_t \right\rangle - \left\|\nabla f(x_t)\right\|_\infty d\left\|\eta_{t-1}A_{t-1} - \eta_t A_t\right\|_1 \left\|g_t\right\|_\infty \\
&\quad \left(By\ H\ddot{o}lder's\ inequality\right) \\
&\ge \left\langle \eta_{t-1} A_{t-1} \nabla f(x_t), g_t \right\rangle - dN^2\mathbf{1}_{t\neq a_{\tilde{a}_t}}\left(\left\|\eta_{t-1}A_{t-1}\right\| - \left\|\eta_t A_t\right\|_1\right) \\
&\quad - dN^2\mathbf{1}_{t=a_{\tilde{a}_t}}\left(\left\|\eta_{t-1}A_{t-1} - \eta_t A_t\right\|_1\right) \tag{26} \\
&\quad \left(Since\ \left\|g_t\right\|_\infty \le N,\ \eta_{t-1} \ge \eta_t > 0,\ A_{t-1} = A_t\ when\ t \neq a_{\tilde{a}_t}\right)
\end{aligned}$$

Perform telescope sum, we have

$$\sum_{t=1}^{T}\left\langle Q_t, g_t\right\rangle \geq \sum_{t=1}^{T}\left\langle \eta_{t-1}A_{t-1}\nabla f(x_t), g_t\right\rangle - dN^2\sum_{k=1}^{\tilde{a}_T-1}\left(\left\|\eta_{a_k}A_{a_k}\right\|_1 - \left\|\eta_{a_{k+1}-1}A_{a_{k+1}-1}\right\|_1\right)$$

$$-dN^2\sum_{k=1}^{\tilde{a}_T}\left\|\eta_{a_k-1}A_{a_k-1} - \eta_{a_k}A_{a_k}\right\|_1 - dN^2\left(\left\|\eta_{a_{\tilde{a}_t}}A_{a_{\tilde{a}_t}}\right\|_1 - \left\|\eta_T A_T\right\|_1\right)$$

$$\geq \sum_{t=1}^{T}\left\langle \eta_{t-1}A_{t-1}\nabla f(x_t), g_t\right\rangle - dN^2\sum_{k=1}^{\tilde{a}_T-1}\left\|\eta_{a_k}A_{a_k}\right\|_1$$

$$-dN^2\sum_{k=1}^{\tilde{a}_T}\left\|\eta_{a_k-1}A_{a_k-1} - \eta_{a_k}A_{a_k}\right\|_1 - dN^2\left\|\eta_{a_{\tilde{a}_t}}A_{a_{\tilde{a}_t}}\right\|_1$$

$$\geq \sum_{t=1}^{T}\left\langle \eta_{t-1}A_{t-1}\nabla f(x_t), g_t\right\rangle - dN^2\sum_{k=1}^{\tilde{a}_T}\left\|\eta_{a_k}A_{a_k}\right\|_1$$

$$-dN^2\sum_{k=1}^{\tilde{a}_T}\left(\left\|\eta_{a_k-1}A_{a_k-1}\right\|_1 + \left\|\eta_{a_k}A_{a_k}\right\|_1\right)$$

$$= \sum_{t=1}^{T}\left\langle \eta_{t-1}A_{t-1}\nabla f(x_t), g_t\right\rangle - 2dN^2\sum_{k=1}^{\tilde{a}_T}\left\|\eta_{a_k}A_{a_k}\right\|_1 - dN^2\sum_{k=1}^{\tilde{a}_T}\left\|\eta_{a_k-1}A_{a_k-1}\right\|_1$$

$$\geq \sum_{t=1}^{T}\left\langle \eta_{t-1}A_{t-1}\nabla f(x_t), g_t\right\rangle - 3dN^2\sum_{k=1}^{\tilde{a}_T}\eta_{a_k-1}C_u \tag{27}$$

Next, we derive an upper bound for $\sum_{t=1}^{T}\left\langle Q_t, g_t\right\rangle$ by deriving an upper-bound for the RHS of equation 25. We derive an upper bound for each part.

$$\left\langle Q_t, m_t\right\rangle = \left\langle \eta_t A_t\nabla f(x_t), m_t\right\rangle = \left\langle \nabla f(x_t), \eta_t A_t m_t\right\rangle$$

$$= \left\langle \nabla f(x_t), x_t - x_{t+1}\right\rangle$$

$$\leq f(x_t) - f(x_{t+1}) + \frac{L}{2}\left\|x_{t+1} - x_t\right\|^2 \tag{28}$$

$$\left(\textit{By L-smoothness of } f\right)$$

Perform telescope sum, we have

$$\sum_{t=1}^{T}\left\langle Q_t, m_t\right\rangle \leq f(x_1) - f(x_{T+1}) + \frac{L}{2}\sum_{t=1}^{T}\left\|\eta_t A_t m_t\right\|^2 \tag{29}$$

$$\left\langle Q_t - Q_{t+1}, m_t\right\rangle = \left\langle \eta_t A_t\nabla f(x_t) - \eta_{t+1}A_{t+1}\nabla f(x_{t+1}), m_t\right\rangle$$

$$= \left\langle \eta_t A_t\nabla f(x_t) - \eta_t A_t\nabla f(x_{t+1}), m_t\right\rangle$$

$$+ \left\langle \eta_t A_t\nabla f(x_{t+1}) - \eta_{t+1}A_{t+1}\nabla f(x_{t+1}), m_t\right\rangle$$

$$= \left\langle \nabla f(x_t) - \nabla f(x_{t+1}), \eta_t A_t m_t\right\rangle + \left\langle (\eta_t A_t - \eta_{t+1}A_{t+1})\nabla f(x_t), m_t\right\rangle$$

$$= \left\langle \nabla f(x_t) - \nabla f(x_{t+1}), x_t - x_{t+1}\right\rangle + \left\langle \nabla f(x_t), (\eta_t A_t - \eta_{t+1}A_{t+1})m_t\right\rangle$$

$$\leq L\left\|x_{t+1} - x_t\right\|^2 + \left\langle \nabla f(x_t), (\eta_t A_t - \eta_{t+1}A_{t+1})m_t\right\rangle$$

$$\left(\textit{By smoothness of } f\right)$$

$$\leq L\Big\|x_{t+1} - x_t\Big\|^2 + \Big\|\nabla f(x_t)\Big\|_\infty d\Big\|\eta_t A_t - \eta_{t+1}A_{t+1}\Big\|_1 \Big\|m_t\Big\|_\infty$$

$$\Big(\textit{By Hölder's inequality}\Big)$$

$$\leq L\Big\|x_{t+1} - x_t\Big\|^2 + dN^2\mathbf{1}_{t+1\neq a_{\tilde a_{t+1}}}\Big(\Big\|\eta_t A_t\Big\|_1 - \Big\|\eta_{t+1}A_{t+1}\Big\|_1\Big)$$

$$+ dN^2\mathbf{1}_{t+1=a_{\tilde a_{t+1}}}\Big(\Big\|\eta_t A_t - \eta_{t+1}A_{t+1}\Big\|_1\Big) \tag{30}$$

$$\Big(\textit{Since } \eta_{t+1} \geq \eta_t > 0, A_{t+1} = A_t \textit{ when } t \neq a_{\tilde a_t}\Big)$$

Perform telescope sum, we have

$$\sum_{t=1}^{T-1}\Big\langle Q_t - Q_{t+1}, m_t\Big\rangle \leq L\sum_{t=1}^{T-1}\Big\|\eta_t A_t m_t\Big\|^2 + dN^2\sum_{k=1}^{\tilde a_T - 1}\Big(\Big\|\eta_{a_k}A_{a_k}\Big\|_1 - \Big\|\eta_{a_{k+1}-1}A_{a_{k+1}-1}\Big\|_1\Big)$$

$$+ dN^2\sum_{k=1}^{\tilde a_T - 1}\Big\|\eta_{a_{k+1}-1}A_{a_{k+1}-1} - \eta_{a_{k+1}}A_{a_{k+1}}\Big\|_1$$

$$+ dN^2\Big(\Big\|\eta_{a_{\tilde a_T}}A_{a_{\tilde a_T}}\Big\|_1 - \Big\|\eta_T A_T\Big\|_1\Big)$$

$$\leq L\sum_{t=1}^{T-1}\Big\|\eta_t A_t m_t\Big\|^2 + dN^2\sum_{k=1}^{\tilde a_T - 1}\Big\|\eta_{a_k}A_{a_k}\Big\|_1$$

$$+ dN^2\sum_{k=1}^{\tilde a_T - 1}\Big(\Big\|\eta_{a_{k+1}-1}A_{a_{k+1}-1}\Big\|_1 + \Big\|\eta_{a_{k+1}}A_{a_{k+1}}\Big\|_1\Big)$$

$$+ dN^2\Big\|\eta_{a_{\tilde a_T}}A_{a_{\tilde a_T}}\Big\|_1$$

$$\leq L\sum_{t=1}^{T-1}\Big\|\eta_t A_t m_t\Big\|^2 + dN^2\sum_{k=1}^{\tilde a_T}\Big\|\eta_{a_k}A_{a_k}\Big\|_1$$

$$+ dN^2\sum_{k=1}^{\tilde a_T - 1}\Big(\Big\|\eta_{a_{k+1}-1}A_{a_{k+1}-1}\Big\|_1 + \Big\|\eta_{a_{k+1}}A_{a_{k+1}}\Big\|_1\Big)$$

$$\leq L\sum_{t=1}^{T-1}\Big\|\eta_t A_t m_t\Big\|^2 + 2dN^2\sum_{k=1}^{\tilde a_T}\Big\|\eta_{a_k}A_{a_k}\Big\|_1$$

$$+ dN^2\sum_{k=1}^{\tilde a_T - 1}\Big\|\eta_{a_{k+1}-1}A_{a_{k+1}-1}\Big\|_1$$

$$\leq L\sum_{t=1}^{T-1}\Big\|\eta_t A_t m_t\Big\|^2 + 3dN^2\sum_{k=1}^{\tilde a_T}\eta_{a_k-1}C_u \tag{31}$$

We also have

$$\Big\langle Q_T, m_T\Big\rangle = \Big\langle \eta_T A_T \nabla f(x_T), m_T\Big\rangle = \Big\langle \nabla f(x_T), \eta_T A_T m_T\Big\rangle$$

$$\leq L\frac{1-\beta_1}{\beta_1}\Big\|\eta_T A_T m_T\Big\|^2 + \frac{\beta_1}{4L(1-\beta_1)}\Big\|\nabla f(x_T)\Big\|^2$$

$$\Big(\textit{By Young's inequality}\Big)$$

$$\leq L\frac{1-\beta_1}{\beta_1}\Big\|\eta_T A_T m_T\Big\|^2 + \frac{\beta_1 d}{4L(1-\beta_1)}N^2 \tag{32}$$

Combine equation 29, equation 31 and equation 32 into equation 25, we have

$$\sum_{t=1}^{T}\Big\langle Q_t, g_t\Big\rangle \leq L\Big\|\eta_T A_T m_T\Big\|^2 + \frac{\beta_1^2 d}{4L(1-\beta_1)^2}N^2$$

$$+ f(x_1) - f(x_{T+1}) + \frac{L}{2} \sum_{t=1}^{T} \left\| \eta_t A_t m_t \right\|^2$$

$$+ \frac{\beta_1}{1-\beta_1} L \sum_{t=1}^{T-1} \left\| \eta_t A_t m_t \right\|^2 + \frac{3\beta_1}{1-\beta_1} dN^2 \sum_{k=1}^{\tilde{a}_T} \eta_{a_k-1} C_u$$

$$\leq f(x_1) - f(x_{T+1}) + \left( \frac{1}{1-\beta_1} + \frac{1}{2} \right) L \sum_{t=1}^{T} \left\| \eta_t A_t m_t \right\|^2$$

$$+ \frac{\beta_1^2 d}{4L(1-\beta_1)^2} N^2 + \frac{3\beta_1}{1-\beta_1} dN^2 \sum_{k=1}^{\tilde{a}_T} \eta_{a_k-1} C_u \qquad (33)$$

Combine equation 27 and equation 33, we have

$$\sum_{t=1}^{T} \left\langle \eta_{t-1} A_{t-1} \nabla f(x_t), g_t \right\rangle - 3dN^2 \sum_{k=1}^{\tilde{a}_t} \eta_{a_k-1} C_u \leq \sum_{t=1}^{T} \left\langle Q_t, g_t \right\rangle$$

$$\leq f(x_1) - f(x_{T+1}) + \left( \frac{1}{1-\beta_1} + \frac{1}{2} \right) L \sum_{t=1}^{T} \left\| \eta_t A_t m_t \right\|^2$$

$$+ \frac{\beta_1^2 d}{4L(1-\beta_1)^2} N^2 + \frac{3\beta_1}{1-\beta_1} dN^2 \sum_{k=1}^{\tilde{a}_T} \eta_{a_k-1} C_u \qquad (34)$$

Hence we have

$$\sum_{t=1}^{T} \left\langle \eta_{t-1} A_{t-1} \nabla f(x_t), g_t \right\rangle \leq f(x_1) - f(x_{T+1}) + \left( \frac{1}{1-\beta_1} + \frac{1}{2} \right) L \sum_{t=1}^{T} \left\| \eta_t A_t m_t \right\|^2$$

$$+ \frac{\beta_1^2 d}{4L(1-\beta_1)^2} N^2 + \frac{3dN^2}{1-\beta_1} \sum_{k=1}^{\tilde{a}_t} \eta_{a_k-1} C_u$$

$$\leq f(x_1) - f^* + \left( \frac{1}{1-\beta_1} + \frac{1}{2} \right) L \eta_0^2 N^2 C_u^2 d \sum_{t=1}^{T} t^{-2p}$$

$$+ \frac{\beta_1^2 d}{4L(1-\beta_1)^2} N^2 + \frac{3dN^2}{1-\beta_1} \eta_0 C_u \sum_{k=1}^{\tilde{a}_t} (a_k - \mathbf{1}_{k\neq 1})^{-p}$$

$$\leq f(x_1) - f^* + \left( \frac{1}{1-\beta_1} + \frac{1}{2} \right) L \eta_0^2 N^2 C_u^2 d \left( 1 + \int_1^T t^{-2p} \, \mathrm{d}t \right)$$

$$+ \frac{\beta_1^2 d}{4L(1-\beta_1)^2} N^2 + \frac{3dN^2}{1-\beta_1} \eta_0 C_u \sum_{k=1}^{\tilde{a}_t} (a_k - \mathbf{1}_{k\neq 1})^{-p}$$

$$\leq f(x_1) - f^* + \left( \frac{1}{1-\beta_1} + \frac{1}{2} \right) L \eta_0^2 N^2 C_u^2 d \int_1^T t^{-2p} \, \mathrm{d}t$$

$$+ \underbrace{\frac{\beta_1^2 d}{4L(1-\beta_1)^2} N^2 + \frac{3dN^2}{1-\beta_1} \eta_0 C_u \sum_{k=1}^{\tilde{a}_t} (a_k - \mathbf{1}_{k\neq 1})^{-p}}_{J}$$

$$+ \underbrace{\left( \frac{1}{1-\beta_1} + \frac{1}{2} \right) L \eta_0^2 N^2 C_u^2 d}_{K}$$

$$\leq f(x_1) - f^* + K \int_1^T t^{-2p} \, \mathrm{d}t + J + K \qquad (35)$$

Take expectations on both sides, we have

$$\sum_{t=1}^{T} \left\langle \eta_{t-1} A_{t-1} \nabla f(x_t), \nabla f(x_t) \right\rangle \leq \mathbb{E} f(x_1) - f^* + K \int_1^T t^{-2p} \, \mathrm{d}t + J + K$$

$$\leq F - f^* + K \int_1^T t^{-2p} \, \mathrm{d}t + J + K \tag{36}$$

Note that we have $\eta_t$ decays monotonically with $t$, hence

$$\sum_{t=1}^{T} \left\langle \eta_{t-1} A_{t-1} \nabla f(x_t), \nabla f(x_t) \right\rangle \geq \eta_0 T^{-p} \sum_{t=1}^{T} \left\langle A_{t-1} \nabla f(x_t), \nabla f(x_t) \right\rangle \tag{37}$$

$$\geq \eta_0 T^{1-p} C_l \frac{1}{T} \sum_{t=1}^{T} \left\| \nabla f(x_t) \right\|^2 \tag{38}$$

Combine equation 36 and equation 38, assume $f$ is upper bounded by $M_f$, we have

$$\frac{1}{T} \sum_{t=1}^{T} \left\| \nabla f(x_t) \right\|^2 \leq \frac{1}{\eta_0 C_l} T^{p-1} \left( F - f^* + K \int_1^T t^{-2p} \, \mathrm{d}t + J + K \right) \tag{39}$$

And it is easy to proved when $a_n = 2^{n-1}$, we have

$$J = \frac{\beta_1^2 d}{4L(1-\beta_1)^2} N^2 + \frac{3dN^2}{1-\beta_1} \eta_0 C_u \sum_{k=1}^{\tilde{a}_t} \left( a_k - \mathbf{1}_{k \neq 1} \right)^{-p} \tag{40}$$

$$\leq \frac{\beta_1^2 d}{4L(1-\beta_1)^2} N^2 + \frac{3dN^2}{1-\beta_1} \eta_0 C_u \sum_{k=1}^{\infty} \left( a_k - \mathbf{1}_{k \neq 1} \right)^{-p} \tag{41}$$

$$= \frac{\beta_1^2 d}{4L(1-\beta_1)^2} N^2 + \frac{3dN^2}{1-\beta_1} \eta_0 C_u \left( 1 + \sum_{k=2}^{\infty} \left( 2^{k-1} - \mathbf{1}_{k \neq 1} \right)^{-p} \right) \tag{42}$$

$$\leq \frac{\beta_1^2 d}{4L(1-\beta_1)^2} N^2 + \frac{3dN^2}{1-\beta_1} \eta_0 C_u \left( 1 + \sum_{k=2}^{\infty} \left( 2^{k-2} \right)^{-p} \right) \tag{43}$$

$$= \frac{\beta_1^2 d}{4L(1-\beta_1)^2} N^2 + \frac{3dN^2}{1-\beta_1} \eta_0 C_u \left( 1 + \sum_{k=1}^{\infty} \left( 2^{-p} \right)^{k-1} \right) \tag{44}$$

$$= \frac{\beta_1^2 d}{4L(1-\beta_1)^2} dN^2 + \frac{3dN^2}{1-\beta_1} \eta_0 C_u \left( 1 + \frac{1}{1 - 2^{-p}} \right) \tag{45}$$

$\square$

