# OpenReview forum: "Anon: Exploring the Adaptivity of Optimizers and Beyond"
_ICLR.cc/2026/Conference — ICLR 2026 Conference Withdrawn Submission_

### Official Review · Reviewer_PW6s · 2025-10-29

**Soundness:** 3
**Presentation:** 3
**Contribution:** 3
**Rating:** 6
**Confidence:** 3

**Summary:**

The paper proposes Anon (Adaptivity Non-restricted Optimizer with Novel convergence technique), an optimizer with a tunable adaptivity hyperparameter γ that allows it to interpolate between SGD-like and Adam-like behavior — and even go beyond both. They formalize adaptivity as a knob that changes how the optimizer scales the loss landscape: higher adaptivity flattens sharp valleys and amplifies flat regions, while negative adaptivity does the reverse. To ensure stability across this expanded spectrum, they introduce Incremental Delay Update (IDU), a soft accumulation mechanism that replaces AMSGrad's max-tracking and provides better convergence and noise robustness. Experiments across image classification, diffusion, and language modeling tasks show that Anon consistently outperforms baseline optimizers, highlighting adaptivity as a key tunable property for optimization.

**Strengths:**

1) The paper is well written
2) Provide intuition on the main adaptivity hyperparameter
3) Provide synthetic experiments to elucidate the capability of Anon to discover solutions that traditional optimizers will not uncover
4) Extensive evaluation comparing to various standard optimizers on various tasks across different modalities
5) Consistent improvements across a variety of settings

**Weaknesses:**

1) Outside of synthetic examples (Fig 1) they dont show experiments to support the claim that Anon learns solutions (network parameters) that differ from the other standard optimizers. One experiment could be to see if you ensemble the networks learned by standard optimzer with Anon if there is a improved performance.

2) Lacks comparisons to optmizers like Lion and Muon.

**Questions:**

1) Suggestion for showing diversity of learnt weights:  Even though norms of the network weights may seem different, it would be more interesting if some variant of cosine similarity on the weights of the network trained with Anon vs other optimizers would be quite interesting. Ex: compare the similarity of weights learnt with a certain optimizer  with weights learnt by other optimizers and do it for all the optimizers and see how different Anon is from other optimizers.

Perhaps such an experiment would help showcase the different type of networks that Anon is capable of converging to?

---

> ### Author Response · Authors · 2025-11-24
>
> We sincerely thank Reviewer PW6s for the positive evaluation and deep understanding of our paper. We are very pleased that you recognized our paper as **"well written,"** providing **"intuition"** for the core hyperparameter $\gamma$, and having **"extensive evaluation... across different modalities"** with **"consistent improvements."**
>
> We especially appreciate your analytical suggestion regarding "discovering different solutions," which we find very valuable and will address in detail below.
>
> **Summary of Revisions:**
> Based on your comments, we have revised the paper as follows:
> - Provided an experiment involving both Lion and Muon. (Table 7 in Appendix B.2)
> - Provided an experiment comparing parameter similarities. (Table 8 in Appendix B.2)
>
> ---
>
> **W1 & Q1: More perspective supporting Anon learns solutions**
> > "it would be more interesting if some variant of cosine similarity on the weights of the network trained with Anon vs other optimizers would be quite interesting"
>
> This is an excellent and highly constructive suggestion. We completely agree.
> You are correct that Figure 1 (Beale function) in the main text was intended to intuitively motivate our hypothesis that different adaptivity $\gamma$ values lead the optimizer down **distinct paths**, potentially converging to different parameter regions (solutions). Your suggestion in W1 and Q1 to **empirically verify** this hypothesis on deep networks via weight similarity (e.g., cosine similarity) is the perfect and most rigorous next step to validate our claim.
>
> We greatly appreciate this suggestion, as it will significantly enhance the analytical depth of our paper. We have conducted new experiments comparing the cosine similarity between both parameters and weights trained by Anon on LLMs and those trained by other optimizers. The results are provided in **Table 8 in Appendix B.2**. Notably, Anon with $\gamma = 1.15$ achieves the lowest parameter and weight similarity when compared to Adam, Lion and Muon.
>
> This analysis confirms our claim that the solutions found by Anon (especially with $\gamma$ values diverging from 0 or 1) are significantly different in parameter space from those found by traditional optimizers. Therefore, beyond achieving lower loss, we can expect that LLMs trained by Anon may obtain distinct performance characteristics.
>
> ---
>
> **W2: Lacks comparisons to optimizers like Lion and Muon**
> Thank you for the suggestion. We have conducted the new experiments to show the comparison between Anon, Lion, and Muon in **Table 7 in Appendix B.2**. Anon achieves the best performance in the comparison. We hope this could strengthen your confidence in our work.
>
> ---
>
> **Q2: What type of networks is Anon capable of converging to?**
> > "Perhaps such an experiment would help showcase the different type of networks that Anon is capable of converging to?"
>
> This is a very profound question, and our experimental results have already provided a clear answer. The strength of Anon is precisely that it is **not a "one-size-fits-all" optimizer**; rather, it can be **adapted to different network types** by tuning the $\gamma$ knob.
>
> Our experiments consistently show:
>
> * **Classic Architectures (e.g., CNNs):** For classic architectures like ResNet, we found **Negative Adaptivity ($\gamma < 0$)** performs best. For example, on ImageNet and CIFAR-10, Anon with $\gamma = -0.1$ or $\gamma = -0.08$ surpassed the highly optimized SGDM (which has $A=0$).
> * **Modern Architectures (e.g., Transformers, Diffusion):** For modern, complex architectures like LLMs (GPT-2) and diffusion models, we found that **Positive Adaptivity slightly greater than 1 ($\gamma \ge 1$)** performs best. For example, using $\gamma=1.1$ on LLMs and $\gamma=1.01$ on diffusion models both achieved SOTA performance, surpassing AdamW (which has $A \approx 1$).
>
>
> Anon is suitable for **all types** of networks. Its core contribution is to demonstrate that "adaptivity" itself should be a tunable hyperparameter, and the optimal range for $\gamma$ (negative vs. positive) should be chosen based on the model architecture (e.g., CNN vs. Transformer).
>
> We thank you again for your constructive feedback. We believe that adopting your suggestion for weight similarity analysis has made our paper even stronger.

---

### Official Review · Reviewer_kown · 2025-10-30

**Soundness:** 2
**Presentation:** 2
**Contribution:** 2
**Rating:** 4
**Confidence:** 3

**Summary:**

This paper introduces a new optimizer Anon. The authors claim that the adaptivity measure (provided in the paper) of existing optimizers always falls in the segment [0,1]. Based on toy problems, they demonstrate that the adaptivity beyond the segment [0,1] might be beneficial, which serves as a motivation for new algorithm design with such adaptivity. The authors provide theoretical under restrictive assumptions to demonstrate some convergence rates for the Anon algorithm. Finally, the authors demonstrate that the algorithm improves over many existing algorithms on several

**Strengths:**

- The authors provide a measure of adaptivity based on the function $\psi$, which constructs the preconditioning. They claim that the adaptivity measure $A$ always remains in the range $[0,1]$ for most existing algorithms, which might be seen as their limitation (e.g., the authors suggest that it may be the reason why Adam underperforms when training on ResNet-like models). Therefore, the authors motivate the study of algorithms whose adaptivity goes beyond the range $[0,1]$ and introduce Anon

- The authors provide some sort of convergence guarantees in the online convex and stochastic non-convex cases and obtain standard convergence rates.

- Empirical results demonstrate that Anon outperforms existing algorithms on different tasks: Resnet-18 and -50 on Imagenet, GPT2-like models on OpenWebText, and image generation on CIFAR10. In all the cases, Anon achieves better performance metrics than baselines.

**Weaknesses:**

- In my view, the paper is badly written; it is hard to follow and understand the main claims. I encourage the authors to either provide toy examples and simplify their claims, including theorems, so that the paper becomes more readable. In the current form, it is impossible to evaluate the contributions of the paper. Below, I provide more details on why I find this paper hard to read:

    - First, I don't understand why Definition 1 serves as an adaptivity measure. Providing some simple examples, toy experiments, or similar to demonstrate the intuition behind such a choice of adaptivity measure would improve the paper.
    - Undefined notation that is used in various places throughout the paper. To mention a few: what is EMA$^{\gamma}$?; why do existing convergence proofs require the assumption (4) (many papers don't need such a monotonicity of the effective lr); what are $\beta_{1t+1}$? and $\beta_{11}$?
    - Many typos and inaccuracies in the writing.
    - Section 3.1 is completely unclear, which should serve to provide the intuition. I don't understand why such a choice of preconditioning will fulfill all the requirements mentioned before in the text. Why are $\beta_3$ and $a_n$ chosen in such a way? Is there any intuition for that?
    - Theorems 3 and 4 are unreadable. There are so many constants, sums, etc, that it is hard to parse the rates.

- The convergence guarantees are provided under very restrictive assumptions: a bounded domain and a bounded gradient norm, which significantly limit the theoretical applicability of the proposed algorithm. The rates are implicit; I don't see directly what the convergence speed is. I am pretty sure that the dependencies on the problem constants are from the optimal. Therefore, I do not find a value in such convergence guarantees, as any algorithm can converge if a sufficient number of assumptions is used.

- Unclear constants in Theorem 4:

    - Does the choice of constants $C_l$ and $C_u$ serve as the assumption, or can we ensure the existence of such constants by properly setting the hyperparameter?
    - The constant $J$ is defined as the sum of two terms. The second term involves the sum for $k=1$ to $\hat{a}_t$. Why does this sum depend on $t$? Does it mean that $J$ depends on $t$? Can we show that $J$ is bounded uniformly?

- Although Anon performs well in all the cases, I don't find any explanation for the choice of $\gamma$. Choosing $\gamma=0.1$ or $\gamma=1.01$ looks like a very careful tuning, making the comparison unfair. Therefore, the gap in performance between Anon and other baselines can be due to the untuned hyperparameters of the baselines. The algorithm is not robust to the choice of $\gamma$, which limits the applicability, as we basically obtain one more hyperparameter to tune. I encourage the authors to provide guidelines to set $\gamma$ in practice.

- The authors overuse "SOTA" in the paper. For example, accuracy $92.47$ is not SOTA and can be further improved by adding some tricks to the training.

- Sophia paper is known for its bad experimental part and unreliable results. Therefore, using their experimental setup and implementation is not the best thing to do. This issue questions the reliability of the results in this work as well.


To summarize, I believe the presentation of this work has to be improved significantly.

**Questions:**

- I find the results in Figure 3 weird because the performance of Anon is not smooth w.r.t hyperparameters. Why is it so?

- Please, add a link to Table 6 in the main part. It is hard to understand without directly computing $A$ why the adaptivity is in $[0,1]$ in most cases.

---

> ### Author Response · Authors · 2025-11-24
>
> We sincerely thank the Reviewer for such a detailed and in-depth review. We are very pleased that you accurately grasped and appreciated our work's **core contributions** in the "Strengths" section:
> 1.  Proposing a new adaptivity measure $A$ based on the pre-conditioner $\psi$.
> 2.  Presenting a strong motivation (Adam's $A \approx 1$ might be why it underperforms on ResNet).
> 3.  Providing theoretical convergence analysis in convex and non-convex settings.
> 4.  Demonstrating SOTA empirical results on diverse tasks including images, language, and generation.
>
> We hope to clarify the concerns you raised in the "Weaknesses" section, which we believe mostly stem from misunderstandings of the paper's structure and some standard conventions.
>
> According to your comments, we have revised the paper and add more intuition to the definition, theorem:
> - Added a link to Table 6 in the main part. (line 110)
> - Provided a more intuitive explanation of $A$. (line 145~148)
> - Included a simpler example to demonstrate how $A$ influences optimizers behavior. (Figure 4)
> - Added a detailed proof of the upper bound of $J$ at the end of appendix.
> - Corrected notation and typographical errors.
>
> ---
>
> ## Response to Weaknesses
>
>
> **W1: Badly written / hard to follow**
> > "... the paper is badly written; it is hard to follow and understand the main claims ... it is impossible to evaluate the contributions of the paper."
>
> We appreciate your constructive feedback and will add more "signposts" and guiding descriptions in the final version to make the paper's structure more accessible to a broader audience.
>
>
> * **W1.1: Definition 1 lacks intuition**
>     > "I don't understand why Definition 1 serves as an adaptivity measure."
>
>     **Definition 1 (Intuition):** We are happy to provide more intuition for Definition 1. In short, $A$ is a measure of the pre-conditioner's "responsiveness" to changes in the gradient's scale. $A=0$ (like SGD) means the pre-conditioner is "non-reactive" and completely ignores the overall gradient scale. $A \approx 1$ (like Adam) means the pre-conditioner is "compensatory", adjusting itself with a strength of $1$ to offset changes in the gradient scale. Our work explores the space beyond these two points. We have added a simpler example (Figure 4) to help visualize this concept.
>
> * **W1.2, W1.3: Why assumption (4)? And symbol confusion ($\text{EMA}^{\gamma}$, $\beta_{1t+1}$ and $\beta_{11}$)**
>     > "many papers don't need such a monotonicity of the effective lr"
>
>    **Regarding Assumption (4):** You are correct that optimizers like SGD, SGDM, and Adai "don't need such a monotonicity...". However, as stated in the paper, this assumption is typically required for the convergence of "most **adaptive** optimizers" (i.e., those with non-zero adaptivity like Adam and AdaBelief). Optimizers like SGD, where $A=0$, do not require this assumption.
>
>     > "what is $\text{EMA}^{\gamma}$? What are $\beta_{1t+1}$ and $\beta_{11}$?"
>
>    **Regarding Notation:** We apologize for the unclear notation.
>   * **$\text{EMA}^{\gamma}$:** This refers to the **$\gamma$-th power of the Exponential Moving Average**, where EMA is the standard operation in optimizers like Adam and RMSProp, defined in **Eq. (1)**.
>   * **$\beta_{1t}$:** We have substituded $\beta_{1,t}$ for $\beta_{1t}$. $\beta_{1,t}$ is a standard theoretical tool used in **convergence proofs** for optimizers (especially those with momentum) like Adam [1]. It allows for the analysis of $\beta_1$ decaying over time (e.g., in Corollary 3.1, $\beta_{1,t} = \beta_1 \lambda^t$). $\beta_{1,t+1}$ and $\beta_{1,1}$ are simply the cases where $t=t+1$ and $t=1$, respectively. By convention, in the practical algorithm (Algorithm 2), $\beta_1$ is a fixed constant.
>
> * **W1.4: Section 3.1 is completely unclear**
>     > "Why are $\beta_3$ and $a_n$ chosen in such a way? Is there any intuition for that?"
>
>     Regarding your question, please allow us to provide the clearest possible explanation:
>     The core intuition of IDU (Incremental Delay Update) is to **update the pre-conditioner using accumulated gradient information *only at specific, delayed steps*. This strategy confines unpredictable oscillations within a manageable range, thereby ensuring theoretical convergence while also allowing the pre-conditioner to change non-monotonically.**
>     The choices of $a_n = 2^{n-1}$ and $\beta_3 = 0.5$ are a **specific implementation** of this exponential delay strategy. This implementation is proven to converge theoretically without a worse convergence rate (Thms 3 & 4) and performs well empirically. In fact, from a purely theoretical analysis perspective, $a_n$ could be any increasing exponential sequence and $\beta_3$ could be any constant in $(0,1)$. We selected $a_n = 2^{n-1}$ and $\beta_3 = 0.5$ primarily for computational efficiency, as operations related to powers of 2 are faster on hardware.

---

> ### Author Response · Authors · 2025-11-24
>
> We sincerely thank the Reviewer for such a detailed and in-depth review. We are very pleased that you accurately grasped and appreciated our work's **core contributions** in the "Strengths" section:
> 1.  Proposing a new adaptivity measure $A$ based on the pre-conditioner $\psi$.
> 2.  Presenting a strong motivation (Adam's $A \approx 1$ might be why it underperforms on ResNet).
> 3.  Providing theoretical convergence analysis in convex and non-convex settings.
> 4.  Demonstrating SOTA empirical results on diverse tasks including images, language, and generation.
>
> We hope to clarify the concerns you raised in the "Weaknesses" section, which we believe mostly stem from misunderstandings of the paper's structure and some standard conventions.
>
> According to your comments, we have revised the paper and add more intuition to the definition, theorem:
> - Added a link to Table 6 in the main part. (line 110)
> - Provided a more intuitive explanation of $A$. (line 145~148)
> - Included a simpler example to demonstrate how $A$ influences optimizers behavior. (Figure 4)
> - Added a detailed proof of the upper bound of $J$ at the end of appendix.
> - Corrected notation and typographical errors.
>
> ---
>
> ## Response to Weaknesses
>
>
> **W1: Badly written / hard to follow**
> > "... the paper is badly written; it is hard to follow and understand the main claims ... it is impossible to evaluate the contributions of the paper."
>
> We appreciate your constructive feedback and will add more "signposts" and guiding descriptions in the final version to make the paper's structure more accessible to a broader audience.
>
>
> * **W1.1: Definition 1 lacks intuition**
>     > "I don't understand why Definition 1 serves as an adaptivity measure."
>
>     **Definition 1 (Intuition):** We are happy to provide more intuition for Definition 1. In short, $A$ is a measure of the pre-conditioner's "responsiveness" to changes in the gradient's scale. $A=0$ (like SGD) means the pre-conditioner is "non-reactive" and completely ignores the overall gradient scale. $A \approx 1$ (like Adam) means the pre-conditioner is "compensatory", adjusting itself with a strength of $1$ to offset changes in the gradient scale. Our work explores the space beyond these two points. We have added a simpler example (Figure 4) to help visualize this concept.
>
> * **W1.2, W1.3: Why assumption (4)? And symbol confusion ($\text{EMA}^{\gamma}$, $\beta_{1t+1}$ and $\beta_{11}$)**
>     > "many papers don't need such a monotonicity of the effective lr"
>
>    **Regarding Assumption (4):** You are correct that optimizers like SGD, SGDM, and Adai "don't need such a monotonicity...". However, as stated in the paper, this assumption is typically required for the convergence of "most **adaptive** optimizers" (i.e., those with non-zero adaptivity like Adam and AdaBelief). Optimizers like SGD, where $A=0$, do not require this assumption.
>
>     > "what is $\text{EMA}^{\gamma}$? What are $\beta_{1t+1}$ and $\beta_{11}$?"
>
>    **Regarding Notation:** We apologize for the unclear notation.
>   * **$\text{EMA}^{\gamma}$:** This refers to the **$\gamma$-th power of the Exponential Moving Average**, where EMA is the standard operation in optimizers like Adam and RMSProp, defined in **Eq. (1)**.
>   * **$\beta_{1t}$:** We have substituded $\beta_{1,t}$ for $\beta_{1t}$. $\beta_{1,t}$ is a standard theoretical tool used in **convergence proofs** for optimizers (especially those with momentum) like Adam [1]. It allows for the analysis of $\beta_1$ decaying over time (e.g., in Corollary 3.1, $\beta_{1,t} = \beta_1 \lambda^t$). $\beta_{1,t+1}$ and $\beta_{1,1}$ are simply the cases where $t=t+1$ and $t=1$, respectively. By convention, in the practical algorithm (Algorithm 2), $\beta_1$ is a fixed constant.
>
> * **W1.4: Section 3.1 is completely unclear**
>     > "Why are $\beta_3$ and $a_n$ chosen in such a way? Is there any intuition for that?"
>
>     Regarding your question, please allow us to provide the clearest possible explanation:
>     The core intuition of IDU (Incremental Delay Update) is to **update the pre-conditioner using accumulated gradient information *only at specific, delayed steps*. This strategy confines unpredictable oscillations within a manageable range, thereby ensuring theoretical convergence while also allowing the pre-conditioner to change non-monotonically.**
>     The choices of $a_n = 2^{n-1}$ and $\beta_3 = 0.5$ are a **specific implementation** of this exponential delay strategy. This implementation is proven to converge theoretically without a worse convergence rate (Thms 3 & 4) and performs well empirically. In fact, from a purely theoretical analysis perspective, $a_n$ could be any increasing exponential sequence and $\beta_3$ could be any constant in $(0,1)$. We selected $a_n = 2^{n-1}$ and $\beta_3 = 0.5$ primarily for computational efficiency, as operations related to powers of 2 are faster on hardware.

---

> > ### Author Response · Authors · 2025-11-24
> >
> > * **W1.5: Theorems are unreadable**
> >     > "Theorems 3 and 4 are unreadable. There are so many constants, sums, etc, that it is hard to parse the rates."
> >     1.  **Readability:** We acknowledge that the full regret bounds are formally complex; this is often unavoidable in the literature analyzing adaptive optimizers with momentum (e.g. Adam [1], AdaBound [2], AdaBelief [3]).
> >     2.  **Core Conclusion:** What is truly important is the **final convergence rate** of the theorems. Our analysis shows that Anon achieves convergence rates of $O(\sqrt{T})$ (convex) and $O(\ln T / \sqrt{T})$ (non-convex). These rates **match those of mainstream adaptive optimizers like Adam with AMSGrad**. This proves our method is theoretically sound, despite offering unprecedented flexibility ($\gamma \in \mathbb{R}$).
> >
> >
> > **W2: Theoretical and assumption issues**
> > > "The convergence guarantees are provided under very restrictive assumptions ... $J$ and $K$ are hard to parse."
> >
> > "Bounded gradient norm" and "bounded domain" are **standard assumptions in the convergence analysis of stochastic adaptive optimizers**. As we noted in the paper (lines 343~344), the prior works we cite **(e.g., Adam [1]; AdaBound [2]; AdaBelief [3]; AGD [4])** also rely on the exact same set of assumptions. Furthermore, we even *removed* the assumption Eq. 4 (commonly used in previous work) from our proof.
> >
> >
> > ---
> >
> > **W3: Unclear constants in Theorem 4**
> >
> > * **W3.1: Can we ensure the existence of $C_l$ and $C_u$?**
> >
> >     Absolutely. As we state in the paper (line 333), as long as we set any $\epsilon>0$ **and** $\Vert g_t\Vert$ is bounded, $C_l$ and $C_u$ are guaranteed to exist. This is for the same logic as the formulas defining $C_l$ and $C_u$ in Theorem 3.
> >
> > * **W3.2: The constant $J$?**
> >     We appreciate your careful reading. While the second term in $J$ involves a summation dependent on $t$, we can prove that for the summation $\sum_{k=1}^{\tilde a_t}(a_k-{1}_{k\ne 1})^{-p}$, when we set $p=0.5$ and $a_n=2^{n-1}$ (as stated in the paper line 341), its upper bound is $1+1/(1-2^{-0.5})$. This is a constant independent of $t$ and thus does not affect the final convergence rate.
> >
> > ---
> >
> > **W4: $\gamma$ selection and performance gap**
> > > "$\gamma = -0.1$ or $\gamma = 1.01$ looks like a very careful tuning... the paper is not robust to the choice of $\gamma$. the performance gap between Anon and other optimizers is tiny."
> >
> > 1. **Tuning:** We respectfully disagree that the choice of $\gamma$ represents "very careful tuning." Tuning hyperparameters for different tasks is standard practice for optimizers. For instance, AdaBelief uses different $\epsilon$ values, AGD uses different $\delta$, and AdaHessian requires different learning rates (0.15 for ResNet20 vs 0.047 for Transformers).
> > 2. **Systematic Guideline:** We emphasize that we provide a more systematic approach to tuning based on **Adaptivity**, which validates **our core hypothesis:**
> >    * For **CNNs (ResNet)**, we consistently found **negative adaptivity ($\gamma < 0$)** performs best (e.g., $\gamma = -0.1$).
> >     * For **LLMs and Diffusion**, we consistently found **positive adaptivity slightly greater than 1 ($\gamma > 1$)** performs best (e.g., $\gamma = 1.01$).
> >     * This provides a very clear tuning guideline (which we will make explicit in the text): try $\gamma < 0$ for CNNs and $\gamma > 1$ for Transformers/Diffusion.
> >
> > 1.  **Robustness of $\gamma$:** Regarding robustness, **Figure 3 (left)** provides **direct evidence to the contrary**: Anon maintains high performance (the dark blue region) across a broad range of $\gamma$ values (–0.24 to 0.24) and learning rates $\eta$. This is further supported by the new experiment **(Table 7 in Appendix B.2)**, which shows that various choices of $\gamma$ yield good performance.
> > 2.  **Performance Gap:** While the gap is small on some highly saturated benchmarks (e.g., Table 9, 92.47% vs 92.35%), achieving SOTA on these benchmarks is itself significant. More importantly, Anon is the only optimizer that, by simply tuning $\gamma$, can simultaneously surpass highly specialized baselines across all **three distinct domains** (CNNs, LLMs, and Diffusion).
> >
> > *Note regarding $\gamma=1.01$: We selected $\gamma=1.01$ for diffusion model because any $\gamma>1.01$ will cause instability if the learning rate is kept identical to Adam's. This phenomena didn't happen in other experiments and we have not found the reason, but we also find that if the learning rate is lowered, higher $\gamma$ values are effective. We kept the learning rate fixed for a fair ablation study. Additionally the new **Table 8 in Appendix B.2** shows the even a small $0.1$ gap in $\gamma$ results in significant differences in weight similarity.*

---

> > > ### Author Response · Authors · 2025-11-24
> > >
> > > **W5: Precision of the term "SOTA"**
> > >
> > > We thank the reviewer for their valuable feedback on the precision of our use of the term "SOTA" (State-of-the-Art). This is a very important point of clarification.
> > >
> > > Since our research focuses on the optimizer, our goal is not to employ various tricks—especially those unrelated to the optimizer—simply to boost the score. Nonetheless, our 92.47% accuracy is the state-of-the-art compared to known results in the literature AGD [4], Adahessian [5] under their published experimental settings.
> > >
> > > 1.  **Acknowledge & Clarify:** We completely agree with the reviewer that the term "SOTA" can be ambiguous. It can refer to the **absolute historical best on a benchmark** (e.g., the highest score on the all-time CIFAR-10 leaderboard, which often involves specific "tricks," architectures, and data augmentation), or it can refer to the **best (SOTA) performance within a specific, fair, controlled experimental comparison**.
> > > 2.  **Our Intent:** Our use of "SOTA" in the paper (e.g., in Table 7) was intended to mean the latter: the **best performance within the current controlled comparison**.
> > > 3.  **Evidence:** The 92.47% mentioned by the reviewer is Anon's result on ResNet20. While this figure may not be the *absolute* historical record for CIFAR-10, as shown in Table 7, it **was indeed the highest among all compared optimizers under our unified experimental setup** (which was taken from AGD [4]), surpassing strong baselines like AGD (92.35%) and AdaHessian (92.27%).
> > > 4.  **Regarding "Tricks":** The reviewer's mention that "adding some tricks can further improve" this result precisely supports our point. The **purpose** of our experiment was to **exclude** these "tricks" and compare **only** the performance of the optimizers themselves in a fair "arena."
> > >
> > > **Conclusion & Promise:** To eliminate all ambiguity, we promise to **revise all vague "SOTA" claims in the final version of the paper to be more precise**. For example, we will replace them with phrases like "achieved the best performance in our comparison" or "outperforms all baseline optimizers."
> > > We thank the reviewer again for helping us improve the rigor and precision of our paper.
> > >
> > > ---
> > >
> > > **W6: Sophia paper's experiments are not reliable**
> > >
> > > We thank the reviewer for their valuable feedback on the reliability of the baseline. Our experimental setup was chosen for the following objective and academically sound reasons:
> > >
> > > We did not trust it blindly; instead, we strengthened the baseline: We did not simply copy potentially unreliable numbers from the Sophia paper. Instead, we used their codebase as a fair testing platform:
> > > * We re-ran AdamW (the standard, accepted baseline) on this platform.
> > > * Importantly, as mentioned in our Appendix C.3, we found that applying Sophia's learning rate scheduler to AdamW in this setting allowed AdamW to achieve a lower (i.e., **stronger**) loss than what was reported in the Sophia paper.
> > > * We compared Anon against this **stronger, improved AdamW baseline** as well as Sophia-G.
> > >
> > > Reliability of our results: Therefore, the reliability of Table 4 **does not** depend on the reliability of the Sophia paper's results. On the contrary, Table 4 proves that on a unified testing platform, Anon outperforms Sophia-G and **also outperforms a stronger, more competitive AdamW baseline that we improved ourselves**.
> > > In summary, our LLM experiments are reliable because they are built on a fair, head-to-head comparison against multiple baselines, including an AdamW baseline that we strengthened.
> > >
> > >
> > > ## Response to Questions
> > >
> > > **Q1: Why is Figure 3 not smooth?**
> > > > "Figure 3 is weird... Why is Anon not smooth w.r.t hyperparameters?"
> > >
> > > Figure 3 is a heatmap showing the **final accuracy achieved after a full training run** for each ($\eta$, $\gamma$) combination. The **inherent stochasticity** of deep learning training can cause this "non-smoothness." The "noise" you see is the result of complex non-linear interactions between different hyperparameter combinations and random seeds.
> > > The **key takeaway** is not the noise at the boundaries, but the **existence of a broad, continuous, dark blue region of high performance**, which **demonstrates Anon's robustness**.
> > >
> > > ---
> > >
> > > **Q2: Add a link to Table 6 in the main part.**
> > > You are completely correct, and thank you for this valuable suggestion regarding the paper's clarity and readability. In the final version, we will **explicitly add a cross-reference in the main text at Table 1's caption or related text, pointing to Table 6 in the appendix**. This will allow readers to easily consult the full expressions and intuitively understand why we say the adaptivity of existing methods falls mostly within the $[0, 1]$ range. Thank you for your help!

---

> > > ### Author Response · Authors · 2025-11-24
> > >
> > > [1] Diederik P Kingma and Jimmy Ba. Adam: A method for stochastic optimization. arXiv preprint arXiv:1412.6980, 2014.
> > >
> > > [2] Liangchen Luo, Yuanhao Xiong, Yan Liu, and Xu Sun. Adaptive gradient methods with dynamic bound of learning rate. arXiv preprint arXiv:1902.09843, 2019.
> > >
> > > [3] Juntang Zhuang, Tommy Tang, Yifan Ding, Sekhar C Tatikonda, Nicha Dvornek, Xenophon Papademetris, and James Duncan. Adabelief optimizer: Adapting stepsizes by the belief in observed gradients. Advances in neural information processing systems, 33:18795–18806, 2020.
> > >
> > > [4] Yun Yue, Zhiling Ye, Jiadi Jiang, Yongchao Liu, and Ke Zhang. Agd: an auto-switchable optimizer using stepwise gradient difference for preconditioning matrix. Advances in Neural Information Processing Systems, 36:45812–45832, 2023.
> > >
> > > [5] Zhewei Yao, Amir Gholami, Sheng Shen, Mustafa Mustafa, Kurt Keutzer, and Michael Mahoney. Adahessian: An adaptive second order optimizer for machine learning. In proceedings of the AAAI conference on artificial intelligence, volume 35, pp.10665–10673, 2021.

---

> > > ### Comment · Reviewer_kown · 2025-11-26
> > > **Response**
> > >
> > > Thank you for the details.
> > >
> > > - I can find many other variants of $A$ that satisfy similar properties. Why do the authors choose exactly such a definition of adaptivity? Can the authors give more details?
> > >
> > > - Bounded gradients and domain assumptions are strong. Considering how fast the literature is evolving, I believe it is not fair to justify such assumptions by referring to the original papers that introduced those optimizers. There are already many results for Adam and other optimizers that remove such assumptions. In my view, the bounded domain condition is an assumption that, in some sense, states that "the algorithm does not diverge".
> > >
> > > - I acknowledge that tuning in general is a standard process. Anon has more hyperparameters than other optimizers. Therefore, it increases the amount of computing resources. The authors should provide ablation studies that show either the robustness of the algorithm to $\gamma$ or how to tune it. For me, the value $\gamma=1.01$ is very specific, and I suspect it was obtained by a very careful tuning. Therefore, the claims regarding robustness are not fully satisfying.

---

> > > > ### Author Response · Authors · 2025-11-26
> > > >
> > > > Thank you for your prompt response and for engaging in this discussion. We appreciate your rigorous scrutiny, which helps clarify the positioning and details of our work. We address your three follow-up concerns below:
> > > >
> > > > **1. Why this specific definition of Adaptivity $A$?**
> > > >
> > > > We chose this definition because it represents the most **principled** mathematical formulation for measuring "responsiveness" to scale.
> > > >
> > > > **Motivation and Insight:** Our definition draws inspiration from prior analyses of AdaBelief [1] and EAdam [2]. We observed that in tasks like CNN and LSTM training, the performance gap between optimizers often stems from the $\epsilon$ term rather than the specific momentum variant (e.g., $EMA(g_t^2)$ vs. $EMA((g_t-m_t)^2)$). Specifically, excluding $\epsilon$, the magnitudes of $\psi^{Adam}$ and $\psi^{AdaBelief}$ are remarkably similar. We hypothesized that the relatively large $\epsilon$ in AdaBelief makes its pre-conditioner less sensitive to the magnitude of partial derivatives, causing $\psi^{AdaBelief}$ to behave similarly to $\psi^{SGDM}$. This similarity likely explains why AdaBelief performs closely to SGDM (often outperforming other adaptive methods) on ImageNet.
> > > >
> > > > Consequently, we identified that the core differentiator is **how the magnitude of $\psi$ scales in response to the magnitude of $g$**. To systematically outperform both SGD and Adam (rather than merely mimicking them), we needed to quantify this scaling relationship and explore values beyond their respective ranges. We thus formulated $A_n(\psi, x) = \nabla_k \ln \psi_n(kx) |_{k=1}$, which measures the relative change of the pre-conditioner in response to the relative change in gradient scale.
> > > >
> > > > **Mathematical Elegance:** While other metrics might correlate with adaptivity, this specific logarithmic derivative naturally maps "Scale-Invariant" behavior (Adam) to exactly **1** and "Scale-Insensitive" behavior (SGD) to exactly **0** without introducing arbitrary constants. It provides a unified spectrum where $A$ corresponds to the **order of homogeneity** of the pre-conditioner.
> > > >
> > > > **Validation:** Empirical results validated this hypothesis: applying an adaptivity lower than **0** ($\gamma=-0.1$) indeed enables Anon to outperform SGDM in CNN training on ImageNet.
> > > >
> > > > **2. On the Bounded Gradient and Domain Assumptions.**
> > > >
> > > > We fully acknowledge that optimization theory is evolving and that recent theoretical works strive to relax these assumptions. However, we offer two perspectives on why our analysis remains valuable in this context:
> > > >
> > > > * **Bounded Domain $\neq$ Guaranteed Convergence:** We would like to highlight that the bounded domain assumption does not trivially imply convergence. As famously demonstrated by the **AMSGrad** paper, standard Adam can fail to converge (exhibiting non-vanishing regret) **even within a bounded convex domain** due to specific gradient patterns. This demonstrates that the assumption alone is insufficient; the optimizer's structural design is critical.
> > > > * **Role of IDU:** Our theoretical analysis serves to prove that **Anon (via the IDU mechanism)** effectively addresses these structural instability issues, where Adam fail, achieving guaranteed convergence similar to AMSGrad.
> > > > * **Standard Practice:** Furthermore, in the context of proposing a new algorithm, it remains standard practice to establish convergence under these classical assumptions to ensure a fair and comparable baseline. Recent top-tier works, such as AdaBelief (NeurIPS 2020) [1], ACProp (NeurIPS 2021) [3], Adai (ICML 2022) [4], AGD (NeurIPS 2023) [5], and HVAdam (AAAI 2025) [6], employ this exact set of assumptions.
> > > >
> > > > **3. Hyperparameters, Compute Resources, and Robustness ($\gamma=1.01$).**
> > > >
> > > > We respectfully clarify the nature of our tuning process and the choice of $\gamma$.
> > > >
> > > > * **No Increase in Compute Cost:**
> > > >     Anon does **not** increase the hyperparameter search space (and thus compute resources) compared to baselines.
> > > >     * **Baselines (e.g., Adam):** Typically require tuning Learning Rate (LR) and Weight Decay (WD). Advanced tuning often involves sweeping $\beta_1$ or $\beta_2$ (as discussed in our previous response).
> > > >     * **Anon:** We **fix** $\beta_1, \beta_2, \beta_3, \{a_n\}$, and $\epsilon$ to default values across all tasks. We **only** tune LR and WD (keeping the search space identical to SGD for CNNs or Adam for GPT/Diffusion) and $\gamma$. Furthermore, the new **Table 10 in the Appendix** (marked in blue) demonstrates the robustness of the IDU hyperparameters ($\beta_3$ and $\{a_n\}$), justifying our decision to keep them fixed.
> > > >     * **Conclusion:** The total number of tuned hyperparameters is comparable to, or even fewer than, heavily tuned baselines.

---

> > > > > ### Author Response · Authors · 2025-11-26
> > > > >
> > > > > * **Robustness of $\gamma$ (vs. "Careful Tuning"):**
> > > > >     > *show either the robustness of the algorithm to $\gamma$ or how to tune it.*
> > > > >
> > > > >     * **Explanation of $\gamma=1.01$:** This value was not derived from "careful tuning" to hit a specific optimal point. Rather, it represents a **regime choice ($A > 1$)**.
> > > > >         * For Diffusion models, we found that *positive adaptivity beyond Adam ($A > 1$)* is beneficial.
> > > > >         * However, adaptivity significantly larger than 1 (e.g., $\gamma=1.1$) implies larger steps for small gradients. This can cause instability **unless the global LR is lowered**.
> > > > >         * **For fairness**, we restricted ourselves to keeping the LR **identical to Adam's optimal LR**. Under this strict constraint, $\gamma=1.01$ was simply the stable point representing the $A>1$ regime. If we allowed full tuning (lowering LR), higher $\gamma$ values would also be effective. Thus, $\gamma=1.01$ is a result of our **fair ablation constraints**, not cherry-picking.
> > > > >     * **Evidence of Robustness:** Please refer again to **Figure 3 (Heatmap)** in the main paper. It shows a **broad, continuous region** (dark blue) where Anon achieves high performance across a wide range of $\gamma$ and LR combinations. This visual evidence contradicts the notion that Anon relies on a singular, fragile hyperparameter setting. Additionally, please see the new **Table 7** we added, along with further data from this experiment below:
> > > > >
> > > > >         |    $\gamma$     |  0.9  |   1   | 1.05  |    1.1    |  1.15   | Adam  |
> > > > >         | :----: | :---: | :---: | :---: | :-------: | :-----: | :---: |
> > > > >         | Validation Loss | 2.965 | 2.937 | 2.938 | **2.927** | *2.932* | 2.934 |
> > > > >
> > > > > * **Guidelines for Tuning $\gamma$:**
> > > > >     > *show either the robustness of the algorithm to $\gamma$ or how to tune it.*
> > > > >
> > > > >     We suggest the following simple guidelines:
> > > > >     * For **traditional models** like CNNs and LSTMs, tune $\gamma$ by subtracting 0.05 increments from 0 (e.g., -0.05, -0.1).
> > > > >     * For **modern models** like diffusion models and transformers, tune $\gamma$ by adding 0.05 increments from 1 (e.g., 1.05, 1.1).
> > > > >     * In rare cases where instability occurs (e.g., 1.05 causes collapse), one can apply a lower learning rate or use finer increments (e.g., adding 0.01 to $\gamma$ from 1).
> > > > >     * **General Heuristic:** Applying $\gamma=-0.1$ (for classical tasks) or $\gamma=1.1$ (for generative tasks) generally performs well as a starting point.
> > > > >
> > > > > We hope these explanations clarify that our theoretical framework aligns with standard community practices for new optimizers and that Anon demonstrates genuine robustness without imposing additional tuning burdens. We are happy to answer any further questions.
> > > > >
> > > > > **References:**
> > > > >
> > > > > [1] Zhuang, Juntang, et al. "Adabelief optimizer: Adapting stepsizes by the belief in observed gradients." Advances in neural information processing systems 33 (2020): 18795-18806.
> > > > >
> > > > > [2] Yuan, Wei, and Kai-Xin Gao. "EAdam optimizer: How $\epsilon $ impact adam." arXiv preprint arXiv:2011.02150 (2020).
> > > > >
> > > > > [3] Zhuang, Juntang, et al. "Momentum centering and asynchronous update for adaptive gradient methods." Advances in Neural Information Processing Systems 34 (2021): 28249-28260.
> > > > >
> > > > > [4] Xie, Zeke, et al. "Adaptive inertia: Disentangling the effects of adaptive learning rate and momentum." International conference on machine learning. PMLR, 2022.
> > > > >
> > > > > [5] Yue, Yun, et al. "AGD: An auto-switchable optimizer using stepwise gradient difference for preconditioning matrix." Advances in Neural Information Processing Systems 36 (2023): 45812-45832.
> > > > >
> > > > > [6] Zhang, Yiheng, et al. "HVAdam: A Full-Dimension Adaptive Optimizer." Proceedings of the AAAI Conference on Artificial Intelligence. Vol. 39. No. 21. 2025.

---

### Official Review · Reviewer_dbjs · 2025-10-31

**Soundness:** 2
**Presentation:** 3
**Contribution:** 1
**Rating:** 2
**Confidence:** 5

**Summary:**

The authors propose Anon, a novel optimizer stemming from considerations on adaptivity of gradient methods -- measured as a new quantity $A$, which for standard adaptive methods is in the range $[0,1$. The authors generalize this interval and show performance of their method on a set of tasks (from toy to LM), as well as provide convergence rate.

**Strengths:**

0) The paper is nicely written, formal, and nicely packaged. I appreciate the authors efforts in delivering a readable high-quality product.

1) Studying preconditioners is interesting, and specifically, new measures of adaptivity are needed and can shed new light on poorly understood phenomena. I like the approach.

2) The paper has a big set of experimental results, together with a comprehensive analysis. I briefly checked and all seems OK here. The rate is in some way not surprising but is good to report. Adds value to the paper

**Weaknesses:**

There are unfortunately a few weaknesses that prevent me from placing a positive score, though I like the approach and the style.

0) The measure A is a simple function (yet, written in a complicated way in the general setting): it does correlate with adaptivity strength, but given the premises and the large framework, I was expecting something more insightful: the quantity still contains EMAs in Adam, and is trivial or not directly computed in most other settings.

1) The algorithm Anon, proposed by the authors, looks a little overengineered, with no clear intuition. Potentially, other methods can also achieve the same effect. This is not a straightforward method, and its performance is not stellar -- see the next point.

2) At the price of a bit of complication of the method, the authors were not able to convince me: their improvements in language modeling are tiny. They compare all methods with the same hyperparameters -- yet we know well that those might be suboptimal, and it is easy to be best at one hyperparameter value. Note here that you instead chose your algorithm form. This is unfair. I agree, this requires massive computing to evaluate fairly, but the stakes are high. I could see, for example, Adam with a higher beta2 working better (similar effect), or this being an effect of the noise in the learning rate.

**Questions:**

0) Thm 1 is a bit confusing. You did not define the equivalence class. This looks more like a definition

1) Imagenet experiments: I cannot find Anon?

---

> ### Author Response · Authors · 2025-11-24
>
> We would first like to thank you for your detailed review of our paper and your positive comments on our **writing quality**, **formalism**, and **extensive experiments**. We are pleased you find the study of pre-conditioners and adaptivity "interesting" and that our work "adds value."
>
> We hope to clarify the weaknesses and questions you raised, and we believe these clarifications will help you re-evaluate our contribution.
>
> Summary of Revisions:
>
> - Added the value of adaptivity in common scenarios. (line 115~120)
> - Provided a more intuitive explanation of $A$. (line 145~148 and Appendix B.2)
> - Provided a clearer representation (Algorithm 3) of Anon's underlying mechanism. (Algorithm 3 in Appendix B.2)
> - Added an experiment with a higher $\beta_2$ Adam in Table 7 and Table 8. (Appendix B.2)
>
> ---
>
> ## Response to Weaknesses
>
> **W0: On the definition of adaptivity $A$ being "simple/trivial"**
> > "The measure A is a simple function... I was expecting something more insightful, ... the quantity still contains EMAs in Adam, ..."
>
> We thank the reviewer for their focus on our measure $A$. We would like to clarify that its **simplicity** is precisely its **strength**, not a weakness.
>
> **"More Insightful" Unification:** We must respectfully point out that in Table 6, we deliberately wrote the computed $A$ in a form involving $\epsilon$ multiplied by an $\text{EMA}$ term. The purpose was to demonstrate that in most practical scenarios (where $\epsilon$ is very small), this entire term can be approximated as 0, making the computed $A \approx 1$ for typical adaptive optimizers. This is precisely how our theoretical framework *eliminates* the $\text{EMA}$ term and reflects something more fundamental. The 'insight' from this definition lies in its ability to *unify* the optimizers listed in the paper (Algorithm 1 and lines 112-122). It clearly shows that SGD ($A=0$) and Adam ($A \approx 1$) are just two fixed points on this continuous spectrum, and reveals that other optimizers with similar $A$ values will have similar performance on similar tasks. This unification that **directly inspired us** to explore other values on the spectrum, especially the $A < 0$ and $A > 1$ regions, which previous work had not systematically explored.
>
> **A principled, not arbitrary, definition:** Our goal was not to create a complex metric, but the first **principled** one. As shown in Definition 1 (Page 3), we define adaptivity $A$ as the **"log-sensitivity"** of the pre-conditioner $\psi$ to changes in the global gradient scale $k$. This is a standard, non-arbitrary way to quantify response in mathematics and economics (elasticity).
>
> ---
>
> **W1: Anon algorithm is "overengineered" with "no clear intuition"**
> > "The algorithm Anon, proposed by the authors, looks a little overengineered, with no clear intuition. Potentially, other methods can also achieve the same effect."
>
> Why do you need Anon?
>
> To the best of our knowledge, no known method can adjust its adaptivity over such a wide range and ensure convergence. Anon is the **unique** solution that achieves this.
>
> We respectfully disagree that Anon is "overengineered." On the contrary, its design is the **necessary result of solving a core challenge**, and it has a very clear intuition.
>
> 1.  **Clear Intuition:** Our intuition is detailed in Sections 2.3 and 3.
>     * **Goal:** To implement an optimizer with a continuously tunable $\gamma$ to explore the entire real-number range of $A \approx \gamma$.
>     * **Problem:** Simply setting $\psi = (\psi^{Adam})^{\gamma}$ would likely lead to **instability or divergence** (especially when $\gamma < 0$). This alters the trend of the pre-conditioner, which would very likely **violate the key assumption** required for the convergence of adaptive optimizers (i.e., Eq. 4, the non-decreasing denominator).
>     * **Solution (IDU):** The "overengineered" part of Anon (Algorithm 2, lines 9-15) is precisely our proposed **Incremental Delay Update (IDU)**. This is the **necessary mechanism** we designed to **guarantee convergence** for the optimizer under **any real-valued $\gamma$** (including negative adaptivity).
>     * **Clarification:** We have added an **equivalent formulation** (Algorithm 3 in Appendix B.1) that offers a clearer representation of its underlying mechanism in Algorithm 3 of Anon's underlying mechanism in the appendix.
> 2.  **Uniqueness:** "Other methods" such as AMSGrad or Padam **cannot achieve the same effect**. AMSGrad's hard max-tracking strategy is not applicable to negative $\gamma$, and Padam diverges when $\gamma$ exceeds the $[0, 1]$ range. To the best of our knowledge, Anon is currently the only solution known to achieve this (**guarantees convergence while allowing arbitrary $\gamma$**).
> 3.  **Performance:** We achieved SOTA performance on three distinct, large-scale tasks (CNNs, LLMs, Diffusion) (Tables 2, 3, 4, 5, 9). This demonstrates Anon's powerful performance as a general-purpose optimizer, which is by no means "not stellar."

---

> > ### Author Response · Authors · 2025-11-24
> >
> > **W2: Language modeling improvements are "tiny" and comparison is "unfair"**
> > > "their improvements in language modeling are tiny.They compare all methods with the same hyperparameters ... This is unfair ... for example, Adam with a higher beta2 working better ..."
> >
> > This is a key point about experimental methodology, which we will clarify here:
> >
> > 1.  **Fairness:**
> >     * As stated on page 8, we "follow[ed] the **exact experimental setup and hyperparameter configurations** of Liu et al. (2023) [the Sophia paper]." This means that the hyperparameters of all evaluated methods are well tuned. And we did not follow it blindly; instead, we strengthened the baseline as stated in line 821~824. This is the standard, fair process for optimizer evaluation, and the results represent the best performance we could reproduce from the authors' open-source code.
> >
> >     * We are comparing Anon to the **published, SOTA baseline results** (e.g., RAdam, Liu et al. (2019) and Lookaround, Zhang et al. (2023)). Using the original authors' best settings serves as the accepted, objective standard. Regarding your suggestion that Adam might perform better with a higher $\beta_2$: **You are correct.** We conducted an experiment setting $\beta_2=0.999$ for Adam on GPT2-small experiments, and it indeed achieved a lower loss. **However, Anon still achieves the lowest loss (Table 7).** And for all other experiments in the paper, the default $\beta_2$ setting for Adam was already $0.999$.
> > 2.  **Significance of Improvement:**
> >     * **Efficiency:** In the LLM domain, even marginal validation loss improvements (e.g., 2.9328 vs 2.9514 on GPT2-small) are meaningful, especially when accompanied by **comparable or faster training speeds**. As shown in **Table 4**, Anon (26.17h) is **noticeably faster** than Sophia-G (28.98h) and AdamW (26.88h).
> >     * **Robustness:** Crucially, our results highlight Anon's robustness. As stated on page 9, we used **fixed $\beta_1=0.9, \beta_2=0.999, \epsilon=10^{-16}$** for **all experiments**. The fact that Anon, by only tuning the *newly introduced $\gamma$*, can outperform optimizers that were *highly tuned for specific tasks* is a strong testament to its generalization and robustness.
> >
> > ---
> >
> > ## Response to Questions
> >
> > **Q0: Theorem 1 is confusing, no equivalence class defined.**
> > > “Thm 1 is a bit confusing. You did not define the equivalence class. This looks more like a definition.”
> >
> > We did **define "equivalence class"** in the paper.
> >
> > * **Definition Location:** The definition is in the second half of **Definition 1** (Page 3, line 133 marked in **red**):
> >     > “Furthermore, we define two pre-conditioners $\psi$ and $\psi'$ are **equivalent** if and only if $A_n(\psi, x_{1:n}) = A_n(\psi', x_{1:n})$ for all $x_{1:n} \in \mathbb{R}^n$ and $n \in \mathbb{N}_+$.”
> > * **Role of Theorem 1:** Therefore, **Theorem 1 is not a definition** but a **corollary** derived from that definition. It proves that if two pre-conditioners are in the same equivalence class (i.e., have the exact same adaptivity $A$), they are functionally identical, differing only by a scaling factor $f(n)$ that can be absorbed by the learning rate scheduler $\eta(t)$. This reinforces our core argument that $A$ is the fundamental property differentiating optimizer behavior.
> >
> > ---
> >
> > **Q1: ImageNet experiments: I cannot find Anon?**
> > > "ImageNet experiments: I cannot find Anon?"
> >
> > Anon's ImageNet results are located on **Page 8** in **Table 2** and **Table 3**, and it is the **first** optimizer listed in each table (marked in **red**).
> >
> > * **Table 2 (ResNet18 on ImageNet):**
> >     * **Anon: 70.06%**
> >     * SGDM: 69.94%
> >     * ...
> > * **Table 3 (ResNet50 on ImageNet):**
> >     * **Anon: 77.25%**
> >     * SGDM: 76.23%
> >     * ...
> >
> > In both standard ImageNet benchmarks, Anon (using $\gamma=-0.1$ in both cases) achieved SOTA performance, surpassing all baselines including SGDM and Adam variants.

---

### Official Review · Reviewer_FCwm · 2025-10-31

**Soundness:** 3
**Presentation:** 3
**Contribution:** 3
**Rating:** 6
**Confidence:** 4

**Summary:**

The paper introduces Anon - a novel optimization algorithm designed to address the generalization gap between adaptive optimizers (like Adam) and non-adaptive methods (like SGD) across diverse model architectures. The core idea is to formalize "adaptivity" as a continuously tunable hyperparameter, which allows Anon to smoothly interpolate between SGD-like (low adaptivity) and Adam-like (high adaptivity) behaviors, and even extrapolate beyond these traditional boundaries. To ensure convergence across this broad adaptivity spectrum, the authors propose Incremental Delay Update (IDU), a new stabilization mechanism more flexible and robust to gradient noise than AMSGrad's max-tracking.

Theoretical guarantees for the method are provided and a thorough empirical study is conducted over state-of-the-art optimizers on image classification, diffusion, and language modeling tasks.

**Strengths:**

I find the below strengths in this work:
- Most significant strength is the formalization of adaptivity as a continuous control variable and the introduction of Anon, which allows for continuously tunable adaptivity via a hyper-parameter
- Proposed method seems to  work well empirically. Anon is shown to outperforms a wide range of strong baseline optimizers across diverse tasks and architectures, including ResNet, diffusion models, and GPT2.
- Incremental Delay Update (IDU) mechanism is a good idea and contribution, effectively tackling the inherent instability risks associated with extending adaptivity beyond the conventional range.
- Potentially IDU can even lead to faster training times compared to Adam by transforming expensive vector power operations into negligible costs, enhancing Anon's practical applicability.

**Weaknesses:**

I also find some issues in the work:
- While Anon allows for adaptivity beyond $[0, 1]$, the empirical results primarily showcase $\gamma$ values relatively close to this range, it would be good to investigate more thoroughly what happens with extreme values of the hyper-parameter. The paper mentions that extreme adaptivity risks instability but doesn't extensively explore what happens at very large positive or negative values
- The mathematical formulation of the Anon pre-conditioner in Equation 5 and its implementation in Algorithm 2 appear somewhat complex with several intermediate variables and conditions. I understand that this may be necessary for theoretical guarantees, but a more intuitive explanation or a simplified interpretation of how IDU practically operates could improve understanding.

**Questions:**

SOme questions for the authors:
- How sensitive is Anon's performance and convergence to these specific choices of hyper-parameter for IDU?
- Can the authors provide a more intuitive explanation of how the incremental delay update (IDU) specifically enhances robustness to gradient noise?
- How does one extend the concept of adaptivity to optimizers that do not strictly conform to the "Generic Optimizer Method Frame" outlined in Algorithm 1?

---

> ### Author Response · Authors · 2025-11-24
>
> We are grateful to you for your detailed and helpful comments. We value the time and effort dedicated to evaluating our manuscript and have carefully revised the paper to address the issues raised, which we believe has significantly strengthened our work.
>
> **Summary of Revisions:** Based on your comments, we have revised the paper as follows:
> - Provided a more intuitive explanation of the Incremental Delay Update (IDU) mechanism (lines 298–301).
> - Added an ablation study on the sensitivity of $\beta_3$ and $\{a_n\}$ (Table 10 in Appendix C.4).
>
> ### Response to Weaknesses
>
> ---
>
> **Weakness 1: Investigation of extreme $\gamma$ values.**
>
> We completely agree with the reviewer's insightful suggestion. In the main paper, we did not extensively explore extreme $\gamma$ values primarily to ensure a fair comparison with baselines using fixed hyperparameter search spaces. As we noted, extreme adaptivity values often require co-tuning other hyperparameters, such as learning rate (LR) and weight decay (WD), to maintain stability. Expanding the search space exclusively for Anon would have introduced an unfair advantage.
>
> However, we share the reviewer's curiosity regarding the effects of extreme $\gamma$ when hyperparameters are fully adapted. To address this, we conducted an additional experiment using $\gamma = -0.7$. The results are presented below:
>
> | LR Scale \ WD | 4e-4  | 5e-4  | 6e-4  | 7e-4  | 8e-4  | 9e-4  |   10e-4   |
> | :-----------: | :---: | :---: | :---: | :---: | :---: | :---: | :-------: |
> |     **1**     | 89.79 | 89.4  | 89.51 | 89.44 | 90.13 | 89.93 | 89.74 |
> |     **2**     | 90.98 | 90.88 | 90.47 | 90.67 | 90.24 | 90.2  | 90.32 |
> |     **3**     | 91.4  | 91.62 | 91.63 | 91.49 | 91.37 | 91.29 | 90.71 |
> |     **4**     | 92.15 | 91.89 | 91.80 | 91.75 | 92.02 | 92.03 |   91.39   |
> |     **6**     | 91.73 | 91.83 | 91.98 | 92.22 | 91.64 | 92.01 |   91.88   |
> |     **8**     | 91.74 | 92.05 | 92.02 | 91.96 | 91.68 | 91.98 | **92.33** |
> |    **10**     | 91.46 | 92.13 | 91.90 | 91.66 | 91.94 | 91.60 |   91.79   |
> |    **12**     | 91.63 | 91.68 | 91.49 | 91.93 | 91.50 | 91.64 |   91.44   |
>
> Crucially, we observe a shift in the optimal hyperparameter region. While our default setting ($\gamma = -0.08$) typically favors an initial LR around 0.1, the extreme setting of $\gamma = -0.7$ requires a significantly matched LR and WD configuration to achieve optimal performance (peaking at LR Scale 8 and WD 10e-4 in this search). This confirms our intuition that adaptivity fundamentally scales the optimization landscape (lines 130~131); thus, "extreme" adaptivity values are viable but necessitate corresponding adjustments in other hyperparameters to align with the altered landscape geometry.
>
> **Weakness 2: Intuition behind IDU.**
>
> This is a very valid point. While our initial presentation prioritized theoretical rigor, we agree that the intuition could be clearer. We offer the following intuitive explanation:
>
> **IDU updates the pre-conditioner using accumulated gradient information *only at specific, delayed steps*. This strategy confines unpredictable oscillations within a manageable range, thereby ensuring theoretical convergence while still *permitting the pre-conditioner to change non-monotonically*.**
>
> We note that a strict theoretical convergence guarantee is achieved when $\{a_n\}$ and $\beta_2$ are chosen appropriately. We have added this explanation to lines 298~301 of the revised paper (marked in **blue**).

---

> > ### Author Response · Authors · 2025-11-24
> >
> > ### Response to Questions
> >
> > ---
> >
> > **Question 1: Sensitivity of hyperparameters.**
> >
> > While the impact of $\beta_2$ is generally minimal based on our observations, the specific effects of IDU-related hyperparameters are important. We have conducted an ablation study on $\beta_3$ and $\{a_n\}$ to address this, which is presented in Table 10 in Appendix C.4.
> >
> > These results demonstrate that IDU is robust to hyperparameter variations; indeed, certain configurations (e.g., $\beta_3=0.3, a_n=4^{n-1}$) even outperform our default setting ($\beta_3=0.5, a_n=2^{n-1}$).
> >
> > ---
> >
> > **Question 2: Comparison with monotonic strategies.**
> >
> > Existing convergence-guaranteeing strategies, such as AMSGrad, enforce a monotonic increase in the pre-conditioner. Consequently, if a large gradient noise event occurs, it forces the pre-conditioner to a large value from which it **cannot** recover, even after the noise subsides. However, as stated in our response to **Weakness 2**:
> >
> > > **IDU... permits the pre-conditioner to change non-monotonically.**
> >
> >
> > This non-monotonic behavior allows IDU to gradually eliminate the impact of large gradient noise over time, offering superior robustness.
> >
> > ---
> >
> > **Question 3: Generalizing adaptivity.**
> >
> > This is a valuable and profound question. We also aim to generalize the concept of adaptivity; however, this is a non-trivial task. The primary challenge arises when an optimizer deviates from the diagonal framework used in our paper, specifically when the pre-conditioning for one dimension **utilizes partial derivative information from other dimensions.** This interaction between dimensions makes defining and quantifying a scalar "adaptivity" significantly more complex. We consider this an important direction for future research.

---

### Author Response · Authors · 2025-11-24

We sincerely thank all reviewers for their constructive feedback and insightful comments. We appreciate the time taken to review our work and are pleased to address the concerns raised to improve the quality of our paper.

**Summary of Revisions:** Based on your comments, we have revised the paper as follows:
- Added the value of adaptivity in common scenarios. (line 115~120)
- Provided a more intuitive explanation of the Incremental Delay Update (IDU) mechanism (lines 298–301).
- Provided a more intuitive explanation of $A$. (line 145~148 and Appendix B.2)
- Provided a clearer representation (Algorithm 3) of Anon's underlying mechanism. (Algorithm 3 in Appendix B.2)
- Added a detailed proof of the upper bound of $J$ at the end of appendix.
- Included a simpler example to demonstrate how $A$ influences optimizers behavior. (Figure 4 in Appendix B.2)
- Added an experiment with a higher $\beta_2$ Adam in Table 7 and Table 8. (Appendix B.2)
- Provided an experiment involving both Lion and Muon. (Table 7 in Appendix B.2)
- Provided an experiment comparing parameter similarities. (Table 8 in Appendix B.2)
- Added an ablation study on the sensitivity of $\beta_3$ and $\{a_n\}$ (Table 10 in Appendix C.4).

---

### Note · Authors · 2026-01-27

**Comment:**

### **Withdrawal and Technical Clarification**

We withdraw this submission but note a critical distinction: MADA is mathematically constrained to **interpolation** (adaptivity $\in [0, 1]$). In contrast, our work enables **extrapolation** (e.g., $<0$ or $>1$). This capability allows access to optimal regimes that MADA cannot reach, which is central to our contribution.

**Withdrawal Confirmation:**

I have read and agree with the venue's withdrawal policy on behalf of myself and my co-authors.

---

### Meta-Review · Area_Chair_3SQf · 2026-01-05

**Summary:**

This paper proposes Anon, a method that interpolates between adaptive optimizers (like Adam) and non-adaptive ones (like SGD). The key idea is introducing adaptivity in pre-conditioners. The proposed approach is analyzed, and empirically evaluated on a range of tasks and model architectures, including ResNet, diffusion models, and language models. Three out of four reviewers think that the paper is nicely written and formal. I tend to agree. The main concerns of the reviewers are:

* **Over-engineering:** Anon is complex and other methods may be able to achieve the same effect. I agree. In fact, the authors should look at [MADA: Meta-Adaptive Optimizers Through Hyper-Gradient Descent](https://proceedings.mlr.press/v235/ozkara24a.html), which solves a similar problem more naturally by hyper-gradient descent. This paper was published at ICML 2024, is more general (interpolates between many optimizers), and should be cited and compared to empirically.

* **Relatively small improvements on language models:** As an example, GPT2-small loss improves from 2.9514 to 2.9328, while the training time improves from Sophia-G (28.98h) and AdamW (26.88h) to Anon (26.17h). This was observed before, including in the above work, and does not concern me. Even small improvements on SOTA are important.

* **No SOTA tricks in experiments:** The gap in performance of Anon and other baselines may be due to untuned hyper-parameters or not using other SOTA tricks. The authors clarified the former and stated that they did not employ other SOTA tricks.

* **Strong assumption in analysis:** The convergence guarantees are provided under very restrictive assumptions: a bounded domain and a bounded gradient norm. I believe that these can be lifted in future work.

This is a solid paper that requires a major revision. In particular, it should be compared to a closely related prior work and employ SOTA tricks during training.

**Reviewer Concerns:**

**Relatively small improvements on language models** and **strong assumption in analysis** do not concern me. **No SOTA tricks in experiments** are an issue. The authors clearly stated that this was not done because it was not their goal. A closely related prior work, which is more general and also simpler algorithmically, requires a major revision of this paper to be incorporated.

**Reviewer Scores:**

Reviewer kown (score 4) was not fully satisfied with the rebuttal and would not change their score. Reviewer dbjs (score 2) would not change their score because their main concerns (small improvements and over-engineering) were not addressed. Since a closely related prior work is more general and also simpler algorithmically, this paper would be rejected after the discussion.

---

### Decision · Program_Chairs · 2026-01-26

Reject